# Noise-Robustness Through Noise: A Framework combining Asymmetric LoRA with Poisoning MoE

**Zhaokun Wang**[1], **Jinyu Guo**[1],*, **Jingwen Pu**[1], **Lingfeng Chen**[1],
**Hongli Pu**[1], **Jie Ou**[1], **Libo Qin**[2], **Wenhong Tian**[1],*
[1]School of Information and Software Engineering,
University of Electronic Science and Technology of China
[2]School of Computer Science and Engineering, Central South University
{guojinyu, tian_wenhong}@uestc.edu.cn

## Abstract

Current parameter-efficient fine-tuning methods for adapting pre-trained language models to downstream tasks are susceptible to interference from noisy data. Conventional noise-handling approaches either rely on laborious data pre-processing or employ model architecture modifications prone to error accumulation. In contrast to existing noise-process paradigms, we propose a noise-robust adaptation method via asymmetric LoRA poisoning experts (LoPE), a novel framework that enhances model robustness to noise only with generated noisy data. Drawing inspiration from the mixture-of-experts architecture, LoPE strategically integrates a dedicated poisoning expert in an asymmetric LoRA configuration. Through a two-stage paradigm, LoPE performs noise injection on the poisoning expert during fine-tuning to enhance its noise discrimination and processing ability. During inference, we selectively mask the dedicated poisoning expert to leverage purified knowledge acquired by normal experts for noise-robust output. Extensive experiments demonstrate that LoPE achieves strong performance and robustness purely through the low-cost noise injection, which completely eliminates the requirement of data cleaning.

## 1 Introduction

Recently, pre-trained language models (PrLMs) have demonstrated remarkable success across various natural language processing tasks (Gao et al., 2025; Bianchi et al., 2024; Guo et al., 2025; Yang et al., 2025). To further enhance model performance on downstream tasks, researchers typically employ domain-specific corpora for targeted fine-tuning of pre-trained models (Li et al., 2024a). However, this mainstream solution critically depends on the quality of training data in both pre-training and fine-tuning stages, though with distinct emphases. Different from the pre-training stage, where models acquire general linguistic knowledge through exposure to large-scale corpora, the fine-tuning stage requires models to adapt to specific downstream tasks using comparatively smaller datasets, thereby imposing stricter requirements on data quality (Mai et al., 2024; Shi and Lipani, 2023).

In downstream NLP applications (Ou et al., 2025), noise poses multiple critical challenges: 1) labeling errors, syntactic irregularities, and extraneous content disrupt the model's ability to capture and learn effective features, leading to unstable training and hindering the establishment of precise, robust knowledge systems (Mao et al., 2023; Xu et al., 2024); 2) noise weakens model generalization when encountering unseen data, impairing flexible knowledge transfer (Ahn et al., 2024); 3) noisy data may introduce biases, causing models to disproportionately emphasize or neglect specific categories,

---

*Corresponding authors.

39th Conference on Neural Information Processing Systems (NeurIPS 2025).

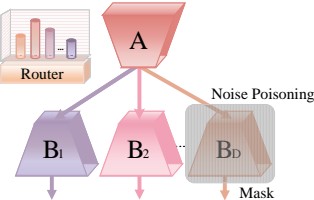

Figure 1: The unique configuration of the asymmetric LoRA architecture in our approach, where the grey area on the right represents the mask for the poisoning expert, thereby eliminating the knowledge affected by the noise learned by the poisoning expert.

compromising fairness and interpretability during fine-tuning stage (Zhao et al., 2024). These issues collectively manifest as drastic performance degradation in downstream tasks.

Recent research on noise can be broadly divided into two types of work. The first type focuses on reconstructing the data before training by cleaning, filtering, or relabeling to construct purified datasets (Ye et al., 2022; Yuan et al., 2024). The second type involves developing dedicated denoising architectures during training (Wu et al., 2023; Zhu et al., 2024). Through the analysis of existing denoising approaches, we identify two primary limitations in current research paradigms: (1) Numerous methods heavily rely on manual intervention or prior assumptions during data pre-processing, requiring noise detection and cleaning before model training; (2) Methods that focus on improving the model architecture during training avoid explicit data cleaning, but still cannot avoid discriminating noise information. In light of this, the effectiveness of these methods is often limited to specific noise types or data distributions. In addition, data processing pipelines not only incur additional computational and annotation expenses but also lead to error propagation.

Fortunately, compared to noise injection identification and processing, noise injection presents a cost-effective and easily automatable alternative that enhances data distribution authenticity through stochastic generation. However, noise addition to training data to improve model robustness seems like a pipe dream, as both clean and noisy samples influence all model parameters, preventing optimal utilization of noise patterns. Therefore, to exploit noise data for beneficial robust effects, a possible approach is to integrate noise injection into a dedicated noise-adaptive module. Subsequently, the module that has learned the noise patterns can be eliminated through complementary set operations, thereby effectively shielding the main network from noise perturbations. We name this strategy "Guided Poisoning".

To achieve this goal, it is necessary to model and separate the noise with dedicated modules to split it. The recent emergence of Mixture-of-Experts (MoE) systems (Shazeer et al., 2017) provides an ideal implementation framework, as their architectural design naturally supports functional specialization through expert combinations. In this context, Low-Rank Adaptation (LoRA) (Hu et al., 2022) within asymmetric MoE architectures offers unique advantages. As shown in Figure 1, the modified MoE-LoRA architectures replace multiple $A$ matrices with a shared $A$ matrix, where each matrix $B$ forms an independent expert. Prior studies (Tian et al., 2024) demonstrate that in such asymmetric LoRA architectures, the shared $A$ matrix typically captures universal knowledge, while individual $B$ matrices adapt to discrepancy knowledge.

Building on these insights, this paper proposes a noise-robust adaptation method via asymmetric **LoRA poisoning experts** (**LoPE**). Specifically, we take one of the matrices $B$ to serve as the poisoning expert within the MoE to route and process noise data as shown in $B_D$ on the right side of Figure 1. During the inference stage, we achieve noise robustness by masking the pathway of the poisoning expert. The overall process consists of two stages: fine-tuning (I-I, I-II) and inference (II).

I-I) In the first stage of fine-tuning, we propose a hybrid noise injection (HyNoIse) approach that combines augmentation at both the input and embedding levels to enhance the original dataset. While freezing other parameters, we train the matrix $A$ and poisoning expert $B_D$ of LoRA with the augmented data. This strengthens the poisoning expert's noise processing capability and leverages the properties matrix $A$ to learn universal knowledge.

I-II) In the second stage of fine-tuning, we focus on the router's ability to guide both clean and noisy data and the normal experts' capacity for learning from clean data. Specifically, we fine-tune

the entire LoRA architecture using original data without noise injection, freezing the poisoning expert $B_D$'s parameters but allowing its participation in forward propagation. Moreover, to further strengthen the integrity of all expert outputs, we propose the Dynamically Compensated Expert Synergy (DyCompEnSate) mechanism to mitigate the impact of removing the poisoning expert during inference.

II) During inference, we only utilize the outputs from normal experts trained on the original data. By masking the poisoning expert, we complete the inference task with nearly noise-free knowledge.

Extensive experiments on benchmark datasets across mainstream NLP tasks demonstrate that our method achieves strong performance without relying on processed clean data, addressing limitations in existing denoising methods. A subsequent series of experiments further validates the fitness between our asymmetric LoRA architecture and noise-robust enhancement. Notably, since LoPE is based on a low-rank parameter-efficient fine-tuning framework, it maintains noise robustness while also achieving high efficiency. The main contributions of this paper are summarized as follows:

1. We propose LoPE, a novel noise-robust adaptation method that utilizes noise injection to handle noise. This approach integrates the noise handling module as an independent expert using the asymmetric LoRA architecture. Combined with multi-stage fine-tuning and masking operations, we enhance the model's robustness to noise only through the generated noisy data.

2. We design a flexible hybrid noise injection strategy, introducing discrete noise at the input level and continuous noise at the embedding level. We also devise a dynamically compensated expert synergy mechanism to compensate for the gap caused by directly masking the poisoning expert.

3. Extensive experiments on multiple mainstream benchmark datasets show that our method achieves competitive performance, which departs from conventional noise-handling paradigms and addresses critical limitations in existing denoising methods.

## 2   Related Work

### 2.1   Research on Data Noise

As data quality continues to play an increasingly crucial role in the performance of deep learning models, handling data noise has become a pressing and critical challenge across various domains. Recently, numerous studies have been conducted to address this issue, aiming to enhance the accuracy and reliability of deep learning models. The primary approach involves constructing denoising models for specific tasks or data structures to learn from the data. (Xu et al., 2024) designed MICL, a contrastive learning module to filter out irrelevant interactions in recommendation systems. (Biester et al., 2024) introduced LLMClean, leveraging generated contextual modeling and rule-based correction for tabular data denoising. (Mao et al., 2023) proposed LeCoRE, which mainly solves the problem of complex search intent understanding in conversational search. These existing methods usually have limitations when dealing with data from different domains, and the trained models are difficult to transfer to other domains. Another common approach involves cleaning, filtering, or relabeling raw data before training to eliminate noisy samples and construct a cleaner dataset. (Feng et al., 2024) introduced an intelligent receiver integrating a pre-denoising network and an LSTM module for denoising preprocessing. (Ji et al., 2024) developed a self-denoising method to refine LLM predictions on noisy inputs. Despite the advancement of existing denoising approaches, they have certain limitations, such as essential noise detection, limitation to data distributions, and error propagation cascades.

### 2.2   Development of LoRA

LLMs have attracted considerable attention due to their powerful natural language processing capabilities (Kenton and Toutanova, 2019). However, the massive parameters of these models make full-parameter tuning prohibitively expensive. Low-Rank Adaptation (LoRA) (Hu et al., 2022) decomposes weight updates into trainable low-rank matrices $(A, B)$, maintaining performance while drastically reducing parameters and computation, establishing it as a key parameter-efficient fine-tuning (PEFT) (Li and Liang, 2021) technique. Recent innovations integrate LoRA with Mixture-of-Experts (MoE) architectures. MoE-LoRA (Dou et al., 2024) synthesizes complementary strengths: MoE's specialized expert networks with gating mechanisms synergize with LoRA's low-rank adap-

tations, enabling cross-domain adaptability. This hybrid approach mitigates multitask interference while preserving parameter efficiency through task-specific adaptor coordination. (Li et al., 2024b) integrated LoRA experts into the feedforward network (FFN) layers of a frozen pre-trained model, improving task adaptability while addressing imbalance issues in MoE models. To address the performance degradation of LoRA in complex corpora while maintaining parameter efficiency, (Tian et al., 2024) developed HydraLoRA, an asymmetric structure that partitions LoRA modules into specialized experts with MoE-based routing, improving efficiency and adaptability in heterogeneous corpora. The asymmetric LoRA structure's unique design enables explicit noise modeling and separates noise-processing modules. This provides a foundation framework for achieving noise robustness through the generated noise.

## 3 Methodology

In this section, we present a comprehensive description of our proposed noise-robust adaptation method via asymmetric LoRA poisoning experts, which encompasses two stages: fine-tuning and inference, as illustrated in Figure 2. We begin by introducing the asymmetric LoRA architecture and the construction of poisoning experts, followed by a sequential explanation of our framework's workflow. Details of each module are given respectively in the remainder of this section.

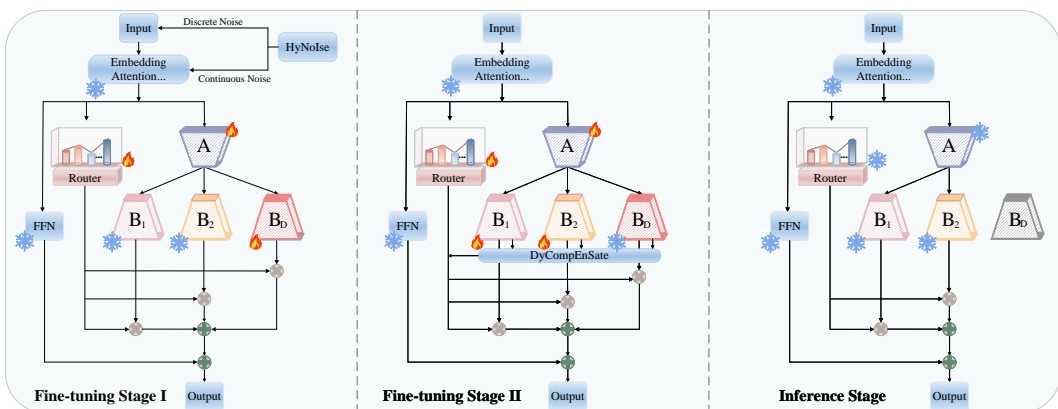

Figure 2: The LoPE pipeline consists of two stages: fine-tuning and inference stages. Fine-tuning Stage I: HyNoIse is to enhance the poisoning expert's noise understanding. Fine-tuning Stage II: Freezing the poisoning expert while fine-tuning the remaining experts and the shared matrix. Inference stage: Masking the poisoning expert with noise-affect knowledge, allowing the remaining experts to generate outputs that are relatively robust to noise.

### 3.1 Asymmetric LoRA architecture

Here we first introduce the asymmetric LoRA architecture (Tian et al., 2024) we employed. This architecture utilizes a centrally shared matrix $A$ and multiple distinct matrices $B_i$, shown as follows: $W = W_0 + \sum_{i=1}^{N} \omega_i B_i A$. Compared to symmetric architectures, asymmetric configurations demonstrate superior advantages in mitigating parameter redundancy and enhancing computational efficiency. By sharing a central matrix $A$, the architecture significantly reduces learnable parameters while maintaining representational capacity, yielding a more lightweight architecture. The implementation of multiple matrices $B_i$ enables flexible assignment of specialized adaptive modules for distinct objectives.

Because of these relatively independent experts, our approach, which involves isolating a dedicated module within the model to separate and process noise, becomes technically feasible. In LoPE, the matrix $B$ is treated as the expert adapter. Furthermore, a randomly selected matrix $B$ is designated as a dedicated poisoning expert, denoted as $B_D$. This expert shares the same structure as the other experts but undergoes a targeted modification of its capabilities through the process described in the following sections, with the aim of identifying and processing noise. Its forward propagation is as follows: $W = W_0 + (\omega_D B_D + \sum_{i=1}^{N-1} \omega_i B_i)A$.

### 3.2 Fine-tuning Stage

#### 3.2.1 Hybrid Noise Injection (HyNoIse)

The original dataset is defined as $S = (x_i, p_i)_{i=1}^M$, where $x_i$ denotes model inputs and $p_i$ their corresponding target labels. To increase the noise ratio in the data, we implement a hybrid data augmentation strategy that integrates both discrete and continuous noise perturbations, with multi-level noise coverage spanning character, token, label, and structural anomalies. To uniformly cover disturbances from all noise types, we adopt an equal-injection strategy. First, discrete noise injection is performed using a function NoiseFunction($\cdot$). Concretely, we adopt three representative and commonly referenced noise strategies that randomly introduce character-level noise (including word order shuffling, noise character insertion, and character deletion) to generate modified inputs, $x_i'$. These transformations enable to simulation of realistic disturbances such as character-level corruption and random truncations. The augmented dataset is then produced as $S' = (x_i', p_i')_{i=1}^M$.

Additionally, we inject continuous noise at the embedding level, influencing both the label and token levels. Inspired by (Jain et al., 2023), let $E \in \mathbb{R}^{b \times l \times d}$ represent the embedding for input sequences, where $b$, $l$, and $d$ correspond to batch size, sequence length, and embedding dimension, respectively. We introduce an attention mask $M \in \mathbb{R}^{b \times l \times 1}$ indicating valid token positions and continuous noise $N \sim \mathcal{U}(-1, 1)^{b \times l \times d}$ from a uniform distribution. The noise-injected embeddings are computed as $E' = E + \alpha N \odot M$, where $\odot$ denotes element-wise multiplication, and $\alpha$ stands for noise ratio. Multiplying the noise with the attention mask element-wise ensures that the noise is applied only to the valid positions of the sequence. The proposed HyNoIse method combines these complementary noise injection strategies to expose the model to diverse noise patterns during fine-tuning.

#### 3.2.2 Fine-tuning Stage I: Specialized Poisoning Expert

In Stage I, we exclusively fine-tune the poisoning expert $B_D$ and shared matrix $A$ using the HyNoIse method from Section 3.1 while keeping the other expert adapters $B_i$ frozen. This stage approach enables $B_D$ focused learning of noise-handling without cross-expert interference. The forward computation during this stage is formalized as:

$$y = W_0 x + (\omega_D B_D + \sum_{i=1}^{N-1} \omega_i f(B_i)) A x, \tag{1}$$

where $\omega_D$ denotes the adaptive weighting coefficient for the poisoning expert, which is part of $\omega_i$, and $f(\cdot)$ stands for freezing parameter operation.

#### 3.2.3 Fine-tuning Stage II: Dynamically Compensated Expert Synergy

In Stage II, we achieve collaboration and complementarity among all expert adapters. To simultaneously enhance the routing mechanism's discriminative capacity for noise patterns and optimize normal experts for clean data processing, we conduct joint fine-tuning of all adapter components while preserving the poisoning expert's specialization. During this phase, we maintain parameter freezing of the $B_D$ from Stage I to keep its capabilities, while updating the remaining expert matrices $B_i$ and shared matrix $A$. The forward computation evolves to:

$$y = W_0 x + (\omega_D f(B_D) + \sum_{i=1}^{N-1} \omega_i B_i) A x. \tag{2}$$

We introduce a router network (gating function) to dynamically allocate expert adapters' contributions. It processes the token representation $x$ through a fully connected layer and softmax to compute expert weights $\omega_i = softmax(W_{gate}^T x)$, where $W_{gate} \in \mathbb{R}^{r \times N}$ denotes the router's parameters. To address the interdependency disruption caused by expert masking in the inference stage, we propose the Dynamically Compensated Expert Synergy (DyCompEnSate) mechanism. Our analysis reveals that joint expert training establishes learned inter-expert correlations, where the direct mask of $B_D$ during inference has the potential to disrupt these dependencies. DyCompEnSate dynamically preserves these relationships through dependency-aware weight compensation. Formally, we construct an expert dependency matrix $\theta \in \mathbb{R}^{N \times N}$, where the element $\theta_{ij}$ quantifies the similarity of output between experts $i$ and $j$. The poisoning expert dependencies $\theta_{iD}$ are computed via output similarity

analysis: $\theta_{iD} = sim(y_i, y_D), \quad i = 1, 2, \ldots, N, i \neq D$, where $sim(\cdot, \cdot)$ denotes cosine similarity metric. These dependencies compensate expert contributions through dynamic weight amplification:

$$y = W_0 x + \beta(\omega_D f(B_D) + \sum_{i=1}^{K}(1 + \theta_{iD})\omega_i B_i)Ax, \tag{3}$$

where $K$ indicates the number of active experts participating in the computation. To maintain output stability, we introduce $\beta$ to adjust the weight parameter normalization.

### 3.3 Inference Stage

During the inference stage, we exclusively employ experts that have acquired purified knowledge during the fine-tuning stage, which naturally leads to masking the poisoning expert while preserving the normal experts. Since the matrices $B_i$ play the role of linear transformations, in the initial stage, we calculate the weighted average of the remaining experts to aggregate their specialized knowledge. Based on this, we perform the parameter-efficient fine-tuning transformation on the input data using the aggregated expert knowledge. In the inference stage, all parameters are frozen, and the contribution of expert adapters is dynamically adjusted based on the input data, thereby achieving the merging of adapters. Specifically, the poisoning expert $B_D$ is masked, such that the system exclusively outputs inference results from the remaining experts trained on clean data to achieve knowledge purification. The result obtained after inference:

$$y = W_0 x + \beta \sum_{i=1}^{K}(1 + \theta_{iD})\omega_i f(B_i)Ax. \tag{4}$$

## 4 Experiments

### 4.1 Datasets, Evaluation Metrics and Compared Baseline Models

**Datasets and Evaluation Metrics.** The dataset used in our experiment is from Alpaca-52K (Taori et al., 2023), which consists of 52K instruction-following data. We used 10% of the dataset for fine-tuning. The dataset spans a wide range of task types, including calculations, question answering, and creation, among others. For evaluation, we first utilized the Massive Multitask Language Understanding (MMLU) (Hendrycks et al., 2020) benchmark, and additionally selected three domains in MMLU encompassing a total of 30 distinct tasks, for in-depth analysis: Natural Sciences and Engineering Technology (NSET), Social Behavior and Humanities (SBH), and History. We also use the GSM8K (Cobbe et al., 2021) math problem set, Question Answering datasets (Physical Interaction QA (Bisk et al., 2020), Social Interaction QA (Sap et al., 2019), and ARC-easy (Clark et al., 2018)). We used different datasets for fine-tuning and evaluating, which effectively mitigates the issue of domain shift. Accuracy (%) is used as the evaluation metric across all datasets. This setup systematically evaluates the model's noise-robust capabilities across general scenarios and specialized domains. A detailed description of these datasets can be found in Appendix A.

**Baselines.** LoPE is a general fine-tuning framework that works across different noise environments and data types, and does not conflict with existing denoising methods. Therefore, we mainly compare it with similar fine-tuning methods. Specifically, we compared LoPE with five baseline PEFT methods on the same datasets: (1) P-Tuning (Liu et al., 2024), (2) Prefix Tuning (Li and Liang, 2021), (3) AdaLoRA (Zhang et al.), (4) LoRA (Hu et al., 2022), and (5) HydraLoRA (Tian et al., 2024). Detailed descriptions of these methods are provided in Appendix B.1.

### 4.2 Implementation Details

We define two modes for the fine-tuning datasets: $\bm{Orig}$, corresponding to the publicly available version discussed in Section 4.1, and $\bm{Nois}$, the version with discrete noise injected. To simulate the real-world noise distributions and create a challenging and impartial noise learning environment across all baseline models, we employed a uniform discrete noise injection strategy same as Section 3.2.1 to generate the $\bm{Nois}$ dataset, introducing a controlled proportion of noise into the original data. Empirical studies (Song et al., 2022) suggest real-world datasets contain 8%-38.5% noise. Given that

Table 1: The comparative performance (%) of different PEFT methods across various datasets. The methods marked with a † are fine-tuned with the $Nois$ datasets, while all subsequent methods are fine-tuned based on $Orig$ datasets. r denotes the rank, #A represents the number of shared matrices, #B represents the number of $B$ matrices involved in the inference stage.

| Methods | MMLU | PIQA | SIQA | GSM8K | ARC-e | NSET | SBH | History | %Param | #A | #B |
|---|---|---|---|---|---|---|---|---|---|---|---|
| HydraLoRA(r=4) | 46.10 | 76.28 | 52.92 | 16.15 | 62.61 | 35.99 | **54.78** | 55.95 | 0.062 | 1 | 3 |
| **LoPE(r=4)** | **46.86** | **76.99** | **53.58** | **17.89** | **64.20** | **36.82** | 54.46 | **56.66** | 0.062 | 1 | 3 |
| | | | | | | | | | | | |
| P-Tuning† | 37.23 | 71.65 | 39.97 | 8.87 | 45.21 | 26.04 | 37.14 | 45.09 | 0.193 | - | - |
| Prefix Tuning† | 37.91 | 71.79 | 40.42 | 9.25 | 43.48 | 27.13 | 38.77 | 44.21 | 0.077 | - | - |
| AdaLoRA(r=2)† | 39.11 | 72.29 | 41.07 | 10.38 | 47.84 | 29.16 | 39.83 | 49.69 | 0.023 | 1 | 1 |
| LoRA(r=2)† | 38.22 | 69.47 | 40.94 | 9.13 | 46.24 | 26.17 | 38.61 | 46.83 | 0.015 | 1 | 1 |
| LoRA(r=4)† | 40.45 | 71.45 | 43.17 | 11.02 | 48.39 | 27.96 | 40.88 | 49.03 | 0.031 | 1 | 1 |
| HydraLoRA(r=2)† | 42.47 | 74.65 | 46.38 | 10.74 | 56.08 | 31.26 | 42.37 | 52.65 | 0.031 | 1 | 3 |
| HydraLoRA(r=4)† | 43.08 | 74.92 | 47.29 | 11.83 | 56.26 | 34.74 | 52.35 | 55.80 | 0.062 | 1 | 3 |
| HydraLoRA(r=8)† | 43.98 | 75.30 | 48.12 | 12.44 | 57.47 | 35.23 | 53.07 | 56.35 | 0.124 | 1 | 3 |
| LoPE(r=2)† | $43.05_{\pm0.28}$ | $75.03_{\pm0.30}$ | $46.76_{\pm0.17}$ | $11.23_{\pm0.37}$ | $58.93_{\pm0.51}$ | $33.36_{\pm0.47}$ | $48.65_{\pm0.32}$ | $54.72_{\pm0.29}$ | 0.031 | 1 | 3 |
| LoPE(r=4)† | $43.76_{\pm0.20}$ | $75.49_{\pm0.41}$ | $48.33_{\pm0.30}$ | $12.72_{\pm0.33}$ | $58.66_{\pm0.46}$ | $34.06_{\pm0.11}$ | $48.98_{\pm0.37}$ | $55.46_{\pm0.18}$ | 0.047 | 1 | 2 |
| LoPE(r=4)† | $44.42_{\pm0.18}$ | $76.28_{\pm0.38}$ | $49.03_{\pm0.41}$ | $13.72_{\pm0.34}$ | $60.49_{\pm0.27}$ | $35.45_{\pm0.33}$ | $52.88_{\pm0.20}$ | $56.84_{\pm0.46}$ | 0.062 | 1 | 3 |
| **LoPE(r=8)†** | $\mathbf{44.82}_{\pm0.24}$ | $\mathbf{76.83}_{\pm0.35}$ | $\mathbf{49.90}_{\pm0.27}$ | $\mathbf{14.31}_{\pm0.38}$ | $\mathbf{62.36}_{\pm0.47}$ | $\mathbf{35.79}_{\pm0.13}$ | $\mathbf{53.71}_{\pm0.32}$ | $\mathbf{58.02}_{\pm0.34}$ | 0.124 | 1 | 3 |

the original dataset already contains some inherent noise, we used a conservative injection rate of 5% to ensure consistency across models and maintain typical noise levels. The HyNoIse ratio $\alpha$ used in our main experiment is also set to 5%, and the number of poisoning experts is set to 1. Detailed descriptions of experiment configurations are provided in Appendix B.2.

## 4.3 Main Results

In this section, we evaluate LoPE's effectiveness through extensive experiments and compare it with several mainstream PEFT methods on LLaMA2-7b (Touvron et al., 2023). Table 1 presents the performance of LoPE alongside the baseline methods. MMLU represents the average accuracy across all tasks. NSET, SBH, and History represent the average accuracy of tasks within each respective set. The specific accuracy for each sub-task can be found in Appendix D.2. We first compared LoPE with HydraLoRA on the $Orig$ datasets mentioned in Section 4.2. In the original environment, LoPE exhibited only a limited improvement across these datasets. Given that the $Orig$ datasets are nearly noise-free with minimal inherent noise, LoPE's noise adaptation capability was not significantly demonstrated. Subsequently, we conducted experiments on the $Nois$ datasets. The results indicate that LoPE exhibits robustness and adaptability across most tasks. Notably, LoPE excelled across three QA tasks, achieving a 4.89% improvement on the ARC-e with a rank of 8. Consistent with our theoretical analysis, LoPE also showed an improvement on the GSM8K task. Moreover, LoPE outperformed baseline methods in the MMLU benchmark (average accuracy of 57 sub-tasks), achieving a 1.34% average improvement over HydraLoRA with a rank of 4. Specifically, as the sub-task of MMLU, NSET, similar to GSM8K, often involves precise computations, LoPE maintained coherent reasoning under noise, whereas traditional methods struggled. Furthermore, LoPE also outperforms other methods in complex tasks such as SBH and History, which involve long texts and logical reasoning. In conclusion, LoPE overcomes the limitations of conventional denoising methods, achieving enhanced robustness through noise injection. Notably, our method maintains the same time complexity of $O(n^2)$ as other fine-tuning methods, ensuring high computational efficiency. In Table 1, we also explore LoPE's sensitivity to parameter configurations by examining the impact of varying the rank and the number of experts, as well as comparing its parameter scale with other methods. Higher ranks generally lead to better performance; However, increasing the rank also incurs higher parameter and computational costs, thus requiring a balance between scalability and effectiveness. In the following experiments, the rank is set to 4 by default, with three normal experts and one poisoning expert.

## 4.4 Ablation Study

### 4.4.1 Ablations on HyNoIse

In this section, we explore the effectiveness of the proposed HyNoIse method. We fine-tune the model on $Nois$ data and set different basic noise ratios (3.5%, 5%, 8%). At the same time, the

ratio of our HyNoIse method is 5%, and the results are shown in Table 2. The results indicate that both discrete noise and continuous noise can improve the method. Our method maintains relatively stable performance when handling different levels of noise. To further verify the effectiveness of our proposed HyNoise method, we conduct ablation studies by selectively removing each type of noise (discrete noise or continuous noise) during injection. As evidenced in Table 2, the robustness of the model is somewhat reduced under single-noise conditions. However, performance with either discrete or continuous noise still surpasses the None type noise injection, thereby highlighting the effectiveness of both noise types. Notably, fine-tuning with discrete-only noise outperforms fine-tuning with continuous-only noise. This can be attributed to the fact that discrete noise directly manipulates the original natural language text, facilitating more effective alignment in the semantic space during the fine-tuning stage II.

Table 2: Average accuracy (%) of LoPE on PIQA and SIQA datasets under different noise conditions. None indicates that no noise is injected during the fine-tuning stage I.

| Noise Type | None | Continuous | Discrete | Hybrid |
|---|---|---|---|---|
| 3.5% Level | 60.90 | 61.23 | 62.07 | **63.31** |
| 5% Level | 59.89 | 60.71 | 61.95 | **62.66** |
| 8% Level | 57.48 | 58.68 | 60.14 | **61.86** |

To further investigate whether LoPE's strong performance depends on the consistency between the noise types in the $Nois$ dataset and the discrete noise in HyNoIse, we conducted additional experiments introducing inconsistent noise injection between the dataset construction and the discrete noise in HyNoIse. The results in Table 3 show that when different operations are applied for LoPE training and dataset construction, all LoPE variants achieve considerable improvements over the baseline HydraLoRA. This demonstrates that LoPE's effectiveness generalizes across different noise categories and does not strongly depend on noise type consistency.

Moreover, the observed performance gains on the $Orig$ dataset (Table 1) further indicate that LoPE is effective and generalizable to naturally occurring noisy data. It is worth noting that HyNoIse also injects continuous noise, which contributes to enhancing the model's robustness under multiple types of noise.

Table 3: Performance comparison of LoPE under **inconsistent noise injection** and HydraLoRA. Each row represents a distinct noise operation. Abbreviations used: NCI (Noise Character Insertion), WOS (Word Order Shuffling), CD (Character Deletion), and WR (Word Replacement), representing four types of noise operations.

| Model | Dataset | HyNoIse | MMLU | PIQA | SIQA | GSM8K |
|---|---|---|---|---|---|---|
| LoPE | Orig + NCI | Continuous + WOS | **44.58** | **75.15** | **50.11** | **14.92** |
| HydraLoRA | Orig + NCI | None | 44.24 | 75.03 | 49.75 | 12.78 |
| LoPE | Orig + WOS | Continuous + CD | **45.09** | **75.84** | **50.55** | **14.64** |
| HydraLoRA | Orig + WOS | None | 44.37 | 75.21 | 49.65 | 12.34 |
| LoPE | Orig + WR | Continuous + NCI | **45.13** | **75.35** | **50.17** | **14.73** |
| HydraLoRA | Orig + WR | None | 44.31 | 75.07 | 49.73 | 12.39 |

We further applied HyNoIse directly to the original HydraLoRA framework (Table 4) and observed a notable performance drop. This is mainly due to the absence of the division of labor between normal experts and poisoning experts, and the multi-stage differentiated training, causing noisy and clean data to equally affect the model from the start. Therefore, LoPE's key contribution is its architecture and multi-stage process, enabling explicit noise filtering and robustness.

### 4.4.2 Impact of Backbone LLMs

To evaluate the impact of backbone architectures on LoPE's performance, we conducted model substitution experiments, with results summarized in Table 5. LoPE consistently demonstrates stable performance improvements across LLaMA2-7b (a decoder-only model), T5-large (a classic encoder-decoder model), as well as Qwen2-7b and Qwen1.5-14b. This result indicates that LoPE's

Table 4: Performance comparison of LoPE and HydraLoRA (with/without HyNoIse) across benchmarks. The methods marked with a † are fine-tuned with the $Nois$ datasets.

| Methods | MMLU | PIQA | SIQA | GSM8K |
|---|---|---|---|---|
| HydraLoRA†(with HyNoIse) | 40.57 | 73.40 | 45.03 | 8.47 |
| HydraLoRA† | 43.08 | 74.92 | 47.29 | 11.83 |
| LoPE† | **44.42** | **76.28** | **49.03** | **13.72** |

performance gains primarily stem from its asymmetric LoRA architecture and two-stage training strategy, which strengthen its generalization across diverse tasks and application scenarios.

Table 5: Performance (%) of different backbone LLMs of our method on SIQA dataset after fine-tuning of $Nois$ datasets (5%).

| Approaches | T5-large | LLaMA2-7b | Qwen2-7b | Qwen1.5-14b |
|---|---|---|---|---|
| HydraLoRA | 33.64 | 47.29 | 66.49 | 78.23 |
| LoPE | **36.37** | **49.03** | **68.99** | **80.02** |

### 4.4.3 Ablation Study on DyCompEnSate

We verified the effectiveness of the proposed DyCompEnSate method. First, we directly masked the output of $B_D$ without introducing the DyCompEnSate mechanism, as shown in Figure 3, when the column number = 0, which indicates that no experts are involved in the DyCompEnSate method, accuracy dropped at the lowest value.

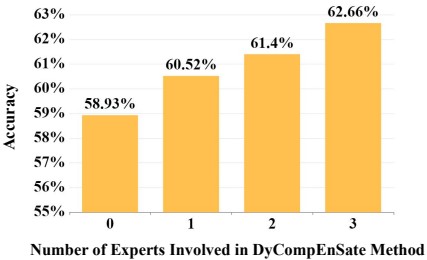

Figure 3: The relationship between the number of experts involved in the DyCompEnSate method and the average accuracy (%) of datasets PIQA and SIQA.

This phenomenon validates our hypothesis: when all experts are jointly trained, there exist significant inter-dependencies among them, and directly eliminating the output of $B_D$ indirectly affects other experts' outputs. With the introduction of DyCompEnSate and the increase in the number of normal experts involved, it dynamically weights the amplification of the contribution of other experts to mitigate the information loss caused by removing $B_D$ in a smooth and adaptive manner.

These results demonstrate the existence of inter-expert dependencies and the effectiveness of our compensation strategy in preserving performance when masking a compromised expert.

## 5 Analysis

### 5.1 Does Higher HyNoise Ratio Enhance Performance?

In Section 4.2, while our baseline experiments employed a 5% HyNoIse ratio, we investigated whether an increased noise ratio during Stage I enhances noise robustness. We evaluated four HyNoIse ratios (3.5%, 5%, 10%, 15%), with comparative results visualized in Figure 4. Contrary to potential expectations, our analysis reveals a non-monotonic relationship between HyNoIse intensity and model robustness. Excessive noise injection (i.e. $\geq 10\%$) degrades the router's discriminative capability for noise patterns, as evidenced by reduced expert specialization. We consider that this was due to overwhelming data distortion, which prevents effective differentiation between clean and

noisy inputs, ultimately compromising expert weight allocation. We also evaluated the performance under different basic noise ratios; additional experimental results are provided in Appendix D.1.

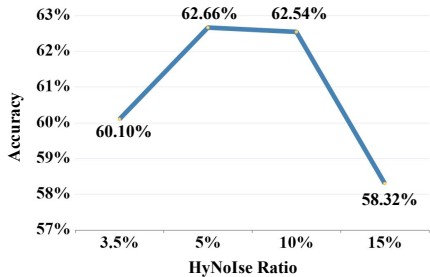

Figure 4: Average accuracy to different HyNoIse ratios in PIQA and SIQA datasets.

## 5.2 Can the Poisoning Expert Truly Accomplish Its Task?

The core of our proposed method is the pioneering introduction of the poisoning expert. In Section 4.3, we investigated the impact of the number of normal experts in LoRA on the experimental results. Here, we discuss the influence of the number of poisoning experts on the results. According to the design, poisoning experts play an auxiliary role. Therefore, their number should be lower than normal experts. We conducted experiments with three different configurations of the number of experts, and the parameter freezing settings remain consistent with Section 3. The experimental results in Table 6 demonstrate that increasing the number of poisoning experts has little impact, as it does not affect the objective of stage I of fine-tuning. These poisoning experts share the functionality previously assigned to a single poisoning expert. During stage II of fine-tuning, after freezing their parameters, the normal experts can learn clean knowledge as usual. When the total number of experts remains constant, changing the number of poisoned experts will have a significant impact on the results, as the normal experts responsible for inference are replaced.

Table 6: Average accuracy of LoPE in PIQA and SIQA datasets with different experts, PE is the number of poisoning experts, NE is the number of normal experts.

| Method | NE=3,PE=1 | NE=3,PE=2 | NE=2,PE=2 | Mask(NE=3,PE=1) | Not Mask(NE=3,PE=1) |
|--------|-----------|-----------|-----------|-----------------|---------------------|
| LoPE   | **62.66** | 62.31     | 60.31     | **62.66**       | 60.75               |

Also, we propose that the involvement of the poisoning expert in the inference process may result in the contamination of knowledge to a catastrophic degree. In light of this, we set up a comparative experiment: Masking vs. Not Masking. The experimental results show that after the first-stage fine-tuning, the poisoning expert intends to specifically adapt and handle the noisy data. If the poisoning expert's output is not masked during inference, allowing it to participate, the model may be affected by the misinformation it learns from the noise, which may affect the reliability of the inference results. This aligns with our original intention for designing the poisoning expert. By masking the poisoning expert that learns from noise-affected data during inference, this approach allows other experts to perform inference using relatively pure knowledge.

## 6 Conclusion

In this paper, we introduce LoPE, a noise-robust adaptation method via asymmetric LoRA poisoning experts, which is specifically designed to address performance degradation caused by noisy datasets. The two-stage paradigm and selective masking mechanism enable the model to enhance its robustness only through generated noise perturbations, as evidenced by our experimental results. This approach effectively overcomes the inherent limitations of conventional noise-handling methods. In future research, we believe the combination of the two-stage fine-tuning paradigm and selective masking mechanism opens a door to a promising area of intractable problem resolution through complement set approaches, and we will explore it for more than noise processing in the future.

## Acknowledgments and Disclosure of Funding

This research is supported by the Sichuan International Science and Technology Innovation Cooperation Project (also known as the Hong Kong, Macao, and Taiwan Science and Technology Innovation Cooperation Project, ID: 2024YFHZ0317), the Chengdu Science and Technology Bureau Project (ID: 2024-YF09-00041-SN), the National Natural Science Foundation of China (Project ID: W2433163), Huawei Funding (Project ID: H04W241592), and the Postdoctoral Fellowship Program (Grade C) of the China Postdoctoral Science Foundation (Grant No. GZC20251053).

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

# Appendix

## A    Dataset

### A.1    Fine-tuning Dataset

Alpaca 52k is a dataset consisting of 52,000 instruction-following data, generated by Stanford University using the self-instruct method by the text-davinci-003. This dataset aims to enhance the model's ability to understand and follow instructions. The dataset covers a wide range of domains and task types, including text generation, reasoning and explanation, transformation and calculation, classification and induction, question answering and suggestions, description and explanation, functionality and logic, as well as text processing and editing. By fine-tuning on these tasks, the model is better equipped to understand and execute various instructions, thereby improving its effectiveness and reliability in real-world applications.

### A.2    Evaluation Dataset

Additionally, to comprehensively evaluate the model's performance across different domains, we employed multiple evaluation benchmarks. To assess general-domain capabilities, we utilized the Massive Multitask Language Understanding (MMLU) benchmark. MMLU spans tasks across multiple fields, enabling a thorough evaluation of the model's language understanding abilities. For our evaluation, we focused on three major categories: Basic Natural Sciences and Engineering Technology (NSET), Social Behavior and Humanities (SBH), and History. The specific tasks are listed in the table 7 below.

Table 7: Task categories and corresponding tasks

| Task Category | Tasks |
| --- | --- |
| NSET | College Biology, College Chemistry, College Computer Science, College Mathematics, College Physics, Electrical Engineering, Abstract Algebra, Astronomy, Machine Learning, Computer Security, High School Chemistry, Elementary Mathematics, High School Physics, High School Computer Science, High School Statistics, High School Biology, High School Mathematics, Conceptual Physics. |
| SBH | Moral Disputes, Moral Scenarios, Professional Psychology, World Religions, High School Psychology, Philosophy, High School Government and Politics, Sociology. |
| History | Prehistory, High School US History, High School World History, High School European History. |

NSET consists of 18 tasks related to science, engineering, technology, and mathematics. This category encompasses foundational disciplines within the natural sciences and engineering fields, including biology, chemistry, computer science, mathematics, physics, electrical engineering, astronomy, and more. These tasks cover biological knowledge, ranging from cell structure and molecular biology to ecology. In chemistry, tasks span from analytical chemistry to organic chemistry, covering fundamental laws (e.g., Newtonian mechanics and principles of chemical reactions). In computer science, research focuses on algorithms, graph theory, recursive methods, and the logic of technical implementation (e.g., algorithm design and circuit systems). Mathematical tasks encompass various branches, such as calculus, combinatorics, and ordinary differential equations. These tasks are highly sensitive to data and require significant specialization and logical reasoning, making them ideal for evaluating the model's precision in computation and reasoning.

SBH includes 8 tasks from fields such as psychology, philosophy, sociology, and political science, focusing on human behavior patterns, moral decision-making, and ethical dilemmas within a societal context. The tasks involve analyzing moral controversies (e.g., freedom of speech, the death penalty, addiction), examining moral situations (e.g., violence, theft), exploring psychological principles behind individual behavior (e.g., personality development, emotional changes), and discussing ethical

dilemmas from a philosophical perspective (e.g., skepticism, utilitarianism). Additionally, the tasks address the impact of social structures on individual behavior, including socialization and inequality in sociology, and government functions and socio-political systems in political science.

The History category consists of 4 tasks related to the development of human history, including Prehistory, American history, world history, and European history. The content spans topics from the early evolution of humans and the origins of civilization to major historical events and transformations in modern and contemporary society. These tasks involve complex logical structures, reflecting the model's depth of reasoning and its ability to transfer knowledge within the social sciences domain.

For evaluating mathematical reasoning ability, we adopted the GSM8K dataset, which includes complex mathematical problems and allows for a quantitative assessment of the model's accuracy and efficiency in solving these problems.

The AI2's Reasoning Challenge (ARC) dataset is a multiple-choice response dataset containing science exam questions for grades three through nine. The dataset is divided into two sections: an easy section (Arc-E) and a hard section (Arc-C) for questions that test the model's ability in scientific reasoning. We used the Arc-E section.

Physical Interaction QA (PIQA) is a dataset designed to test a model's ability to predict how objects interact in the physical world. The questions are typically based on everyday physical scenarios.

Social Interaction QA (SIQA) is a question-answering benchmark designed to assess a model's ability to reason about social commonsense intelligence. SIQA focuses on inferring human actions and their social implications. The behaviors in SIQA span a wide range of social contexts, with answer choices including both human-crafted answers and adversarially filtered machine-generated options. Social IQa contains over 37,000 QA pairs, used to evaluate a model's ability to reason about the social implications of everyday events and situations.

Through this multidimensional evaluation, we assessed the proposed method's adaptability to both general and domain-specific tasks comprehensively, offering a thorough examination of the model's capabilities across diverse multi-task settings and providing solid empirical support for future research.

## B  Baseline, Configurations and Backbone LLMs

### B.1  Baseline

1. P-Tuning (Liu et al., 2024): P-Tuning employs a prompt encoder to automatically optimize continuous prompt embeddings, avoiding the limitations of manually designing discrete prompts. By flexibly inserting anchor tokens into the input sequence, P-tuning can effectively improve the alignment accuracy of the semantic space, achieving better task adaptation performance.

2. Prefix Tuning (Li and Liang, 2021): This method injects trainable prefix parameter matrices at the input of each layer of the Transformer. Through a hierarchical representation rectification mechanism, the prefix parameter matrices can guide the model output while keeping the backbone network parameters frozen. This approach allows for efficient adaptation to new tasks while retaining the pre-trained knowledge.

3. AdLoRA (Zhang et al.): Adaptive LoRA is based on a dynamic allocation algorithm that assesses parameter importance. It applies high-rank decomposition configurations to critical weight matrices and employs low-resource allocation strategies for non-core modules. This breaks the traditional paradigm of uniform parameter allocation, adaptively adjusting parameters for different parts to reduce computational and storage overhead while maintaining performance.

4. LoRA (Hu et al., 2022): LoRA (Low-Rank Adaptation) is an efficient fine-tuning technique for adapting pre-trained language models to downstream tasks. It aims to reduce the computational and storage overhead while maintaining the performance of the adapted model. The core idea of LoRA is to introduce low-rank update matrices to the pre-trained model. Instead of fine-tuning all the parameters of the model, LoRA keeps the original pre-trained weights frozen and injects trainable low-rank matrices into each layer of the model.

5.HydraLoRA (Tian et al., 2024): HydraLoRA is a parameter-efficient fine-tuning method based on the LoRA mechanism, which aims to improve the performance of large-scale language models (LLMs) in multi-task learning and cross-domain adaptation. This method involves the decomposition

of the LoRA module into multiple independent expert modules, thereby enhancing the original symmetrical LoRA structure into a single A and multi-B LoRA structure. This modification enables the model to respond to varying task requirements with greater flexibility.

## B.2 Experiment Configurations

The experiment configurations are as follows: learning rate is 0.0002, random seed is 614, and epoch is 1. The hardware and software configurations used in our experiments are as follows.

CPU: Intel(R) Xeon(R) Platinum 8468V, 800MHZ, 48cores; GPU: NVIDIA TESLA H800 80 GB; Operating system: Ubuntu 20.04; Deep learning framework: Pytorch 1.13.1.

## B.3 Backbone LLMs

1. T5-large (Raffel et al., 2020): T5-large is a pre trained language model based on the Transformer architecture, which has wide applications in the field of natural language processing. T5 stands for "Text-to-Text Transfer Transformer", which converts all NLP tasks into text-to-text format for processing. T5-large has over 7.7 million parameters, and compared to smaller T5 model variants, it is able to capture more complex patterns and semantic relationships in text. It performs well in tasks such as text generation, text classification, question answering systems, machine translation, etc. It can generate smooth and natural text, accurately understand context, and demonstrate good adaptability and universality in multiple language environments.

2. LLaMA2-7b: LLaMA2-7b is a large-scale language model launched by Meta, with approximately 7 billion parameters. It has excellent language comprehension and generation capabilities, and can handle various natural language tasks such as text generation, question answering, translation, summarization, etc. It performs well in multiple languages and can be widely applied in various scenarios, providing users with convenient text processing services. Its open-source and free commercial license characteristics also promote its application and research in many fields.

3. Qwen2-7b (qwe, 2024): Qwen2-7b is an open-source model launched by the Alibaba Cloud Tongyi Qianwen team, which supports multimodal input of text, images, and videos. It has 27 language processing capabilities and 32K long context understanding, and performs better than models of the same scale in areas such as code generation and mathematical reasoning. Adopting efficient group query attention (GQA) technology, adapted to various scenarios from mobile to server, and open sourced under Apache 2.0 protocol, it is commercially available for free.

4. Qwen1.5-14b (Bai et al., 2023): Qwen1.5-14b is a language model series including decoder language models of different model sizes. For each size, we release the base language model and the aligned chat model. It is based on the Transformer architecture with SwiGLU activation, attention QKV bias, group query attention, mixture of sliding window attention and full attention, etc.

# C Limitations

Our study has several limitations. The reliance on LoRA's framework somewhat limits its extensibility. Additionally, during the first stage of fine-tuning, the introduction of discrete noise does not specifically consider the distribution of noisy data, which, to a certain extent, increases the uncertainty. This presents a potential direction for future exploration, where more noise modeling techniques could be developed.

# D Additional Experiments

## D.1 Robustness Analysis under Varying Basic Noise Ratios

To investigate the robustness of LoPE when fine-tuning on datasets with higher inherent noise, we systematically evaluated basic noise ratios of 3.5%, 5%, 8%, 10%, 20%, and 30%. As shown in Table 8, although overall performance gradually declines as the noise level increases, it does not fully collapse, demonstrating that LoPE effectively delays performance degradation under high-noise conditions. This behavior can be attributed to several factors. First, the overall quality of fine-tuning datasets impacts final inference performance. When the noise reaches a certain level, it

Table 8: Performance under different noise ratios (transposed version).

| Dataset / Noise Ratio | 3.5% | 5% | 8% | 10% | 20% | 30% | 30% (HydraLoRA) |
|---|---|---|---|---|---|---|---|
| MMLU | 44.59 | 44.42 | 44.15 | 43.81 | 42.35 | 40.78 | 37.96 |
| PIQA+SIQA | 63.31 | 62.66 | 61.86 | 61.27 | 60.14 | 59.50 | 56.83 |
| GSM8K | 15.31 | 13.72 | 11.83 | 9.25 | 8.04 | 7.60 | 7.12 |

blurs the boundary between clean data and noise, and fine-tuning becomes harmful, highlighting the importance of fine-tuning data quality. As a result, for tasks requiring high precision, such as mathematics datasets, the rate of accuracy decline actually slows at higher noise levels. This may be because mathematical pattern-related content in the fine-tuning dataset has already been corrupted, resulting in less impact on the model itself, and the model may no longer recognize that it is learning mathematics-related knowledge.

### D.2 Specific accuracy for sub-task

The specific accuracy for each task can be found in Table 9, Table 10, Table 11, Table 12, Table 13, and Table 14.

Table 9: Performance across each specific task in NSET for LoPE and HydraLoRA ($Orig$).

| Task | Model | |
|---|---|---|
| NSET | LoPE | HydraLoRA |
| College Biology | 44.44 | 43.75 |
| College Chemistry | 32.00 | 30.00 |
| College Computer Science | 37.00 | 37.00 |
| College Mathematics | 38.00 | 34.00 |
| College Physics | 16.67 | 18.63 |
| Electrical Engineering | 42.76 | 45.52 |
| Astronomy | 40.79 | 42.76 |
| Abstract Algebra | 32.00 | 30.00 |
| Machine Learning | 38.39 | 36.61 |
| Computer Security | 58.00 | 58.00 |
| High School Chemistry | 35.96 | 32.51 |
| Elementary Mathematics | 29.89 | 27.51 |
| High School Physics | 28.48 | 29.14 |
| High School Computer Science | 37.00 | 37.00 |
| High School Statistics | 30.56 | 25.00 |
| High School Biology | 47.42 | 49.35 |
| High School Mathematics | 29.26 | 28.89 |
| Conceptual Physics | 44.26 | 42.13 |

Table 10: Performance across each specific task in SBH for LoPE and HydraLoRA($Orig$).

| Task | Model | |
|---|---|---|
| SBH | LoPE | HydraLoRA |
| Moral Scenarios | 27.71 | 24.69 |
| Professional Psychology | 43.14 | 43.63 |
| World Religions | 66.67 | 66.08 |
| Philosophy | 59.49 | 58.84 |
| Moral Disputes | 49.13 | 51.16 |
| High School Government and Politics | 66.84 | 68.39 |
| Sociology | 61.19 | 64.18 |
| High School Psychology | 61.47 | 61.28 |

Table 11: Performance across each specific task in History for LoPE and HydraLoRA($Orig$).

| Task | Model | |
|---|---|---|
| History | LoPE | HydraLoRA |
| High European History | 60.61 | 63.64 |
| Prehistory | 47.22 | 47.22 |
| High School World History | 62.45 | 59.49 |
| High School US History | 56.37 | 53.43 |

Table 12: Performance across each specific task in NSET for LoPE and HydraLoRA($Nois$).

| Task | Model | |
|---|---|---|
| NSET | LoPE | HydraLoRA |
| College Biology | 45.14 | 43.75 |
| College Chemistry | 31.00 | 29.00 |
| College Computer Science | 38.00 | 33.00 |
| College Mathematics | 33.00 | 40.00 |
| College Physics | 17.65 | 15.69 |
| Electrical Engineering | 38.62 | 33.79 |
| Astronomy | 42.11 | 38.16 |
| Abstract Algebra | 32.00 | 34.00 |
| Machine Learning | 38.39 | 37.50 |
| Computer Security | 53.00 | 52.00 |
| High School Chemistry | 33.50 | 33.00 |
| Elementary Mathematics | 26.72 | 27.51 |
| High School Physics | 31.13 | 27.81 |
| High School Computer Science | 41.00 | 38.00 |
| High School Statistics | 21.30 | 23.61 |
| High School Biology | 46.13 | 47.10 |
| High School Mathematics | 28.52 | 29.63 |
| Conceptual Physics | 40.85 | 41.70 |

Table 13: Performance across each specific task in SBH for LoPE and HydraLoRA($Nois$).

| Task | Model | |
|---|---|---|
| SBH | LoPE | HydraLoRA |
| Moral Scenarios | 26.03 | 25.92 |
| Professional Psychology | 44.28 | 44.12 |
| World Religions | 63.74 | 64.33 |
| Philosophy | 56.27 | 56.91 |
| Moral Disputes | 48.84 | 48.55 |
| High School Government and Politics | 65.28 | 64.25 |
| Sociology | 58.21 | 56.22 |
| High School Psychology | 60.37 | 58.53 |

Table 14: Performance across each specific task in History for LoPE and HydraLoRA($Nois$).

| Task | Model | |
|---|---|---|
| History | LoPE | HydraLoRA |
| High European History | 63.61 | 61.21 |
| Prehistory | 49.38 | 47.53 |
| High School World History | 59.92 | 59.07 |
| High School US History | 54.41 | 55.39 |

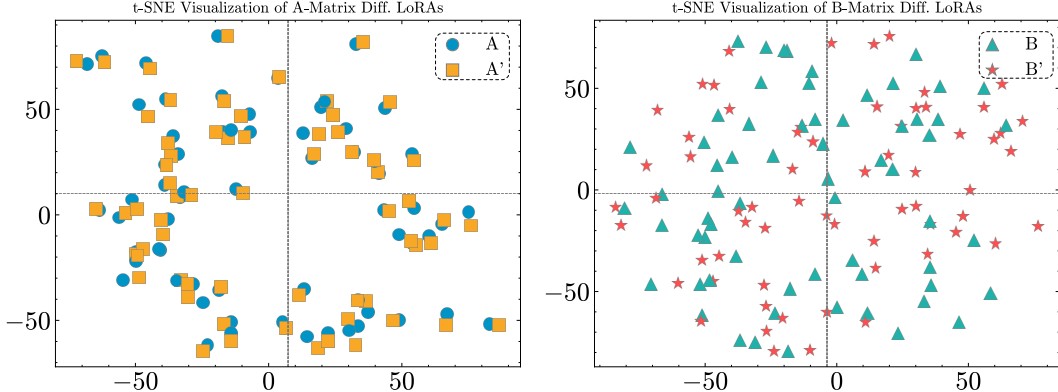

Figure 5: Comparison of the average parameter changes of matrices $A$ and $B$ during first-stage fine-tuning, with and without HyNoIse. Specifically, $A$ and $B$ correspond to the matrix $A$ and poisoning expert $B_D$ trained without HyNoIse, while $A'$ and $B'$ represent the respective matrix $A$ and poisoning expert $B_D$ trained with HyNoIse injected.

### D.3 Matrix Parameter Visualization

We hypothesize that matrix A captures fundamental language-environment knowledge, which is both basic and universal, and is minimally affected by noise (a dataset with added noise can be considered analogous to a dataset from a new domain). To investigate this, we conducted Stage I training with and without HyNoIse data, observing the changes in matrix A parameters to assess whether harmful noise is absorbed into matrix A. Figure 5 shows that the average parameter change of matrix A, with or without HyNoIse, is much smaller than that of matrix B under the same operations. This finding suggests that matrix A primarily encodes universal contextual features present across all data, while the influence of noise on matrix A is substantially smaller than on matrix B.

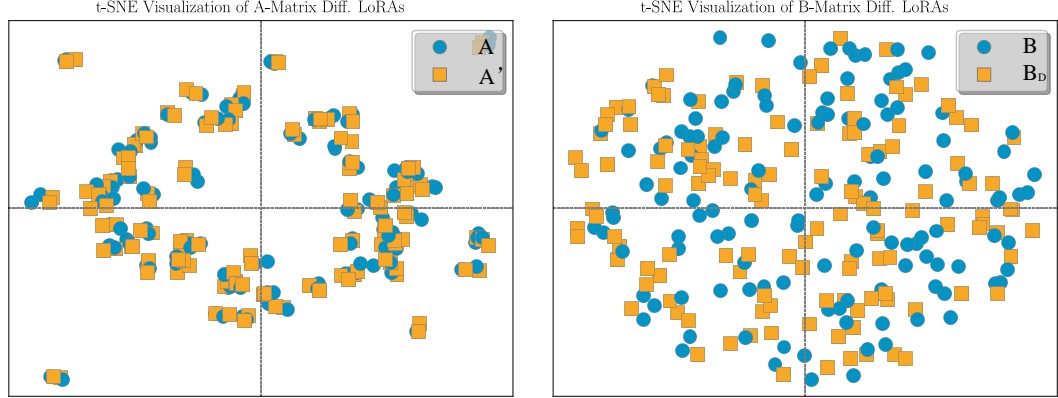

Figure 6: $A$ denotes the parameter visualization of matrix $A$ during the first-stage fine-tuning, while $A'$ represents the updated matrix $A$ in the second stage. $B$ indicates a randomly selected normal expert in the second-stage fine-tuning, and $B_D$ corresponds to the poisoning expert from the first stage. This visualization highlights the parameter evolution of clean and poisoned experts across different training phases.

We further visualize the parameter distributions of matrices A and B across the two-stage fine-tuning process. As shown in Figure 6, the position of matrix A in the visualization space changes very little throughout training. This observation that matrix A primarily encodes general, universal knowledge. In contrast, we observe significant positional shifts between a randomly selected normal expert matrix B and the poisoning expert matrix $B_D$ across the two stages. We attribute this phenomenon to matrix

B capturing task-specific knowledge, while $B_D$ absorbs noise-contaminated patterns, fundamentally differing from the clean representations encoded by other experts.

## D.4 Evaluation against a Task-Specific Denoising Baseline

We have also included a dedicated comparison experiment against a task-specific denoising model to further assess LoPE's generality in downstream applications. Specifically, we investigated the representative denoising method LeCoRE, which is designed for conversational search tasks. Following its evaluation protocol, we conducted experiments under comparable settings. The results are summarized in Table 15.

Table 15: Performance comparison between LoPE and the task-specific denoising model LeCoRE.

| Method | MRR | NDCG@3 | R@10 | R@100 |
|--------|------|--------|------|-------|
| LeCoRE | 51.1 | 48.5 | 73.9 | 89.7 |
| LoPE | **54.7** | **50.1** | **75.6** | **91.3** |

