# OpenReview forum: "Noise-Robustness Through Noise: A Framework combining Asymmetric LoRA with Poisoning MoE"
_NeurIPS.cc/2025/Conference — NeurIPS 2025 poster_

### Official Review · Reviewer_CYJM · 2025-07-01

**Clarity:** 2
**Significance:** 3
**Originality:** 2
**Rating:** 4
**Confidence:** 3

**Summary:**

This work introduces LoPE, a novel fine-tuning framework that strengthens the robustness of LLMs to noisy data. LoPE employs a dedicated "poisoning expert" within an asymmetric LoRA architecture. Through a two-stage fine-tuning process with hybrid noise injection and expert masking during inference, LoPE effectively minimizes the impact of noise without requiring data cleaning or architectural modifications, achieving strong performance.

**Questions:**

1. The role of the shared matrix A remains somewhat unclear, particularly regarding the rationale for fine-tuning it in both Stage I and Stage II. It would be helpful if the authors could clarify the motivation behind this design choice and provide ablation studies to highlight the impact of updating A across both stages.

**Ethical Concerns:**

["NO or VERY MINOR ethics concerns only"]

**Final Justification:**

Thanks for your detailed reply. At this stage I think my concerns are addressed except minor points.

**Limitations:**

Yes.

**Quality:**

3

**Strengths And Weaknesses:**

Pros:
LoPE leverages the HyNoIse and DyCompEnSate components to isolate noise handling to specific experts and mask it during inference without data cleanup. Meanwhile, it is evaluated on different downstream tasks, highlighting its efficiency and robustness

Cons:
1. Sharing and fine-tuning the A-matrix in both Stage I and Stage II raises the question of whether harmful information learned during the first stage (from noisy data) might be absorbed into the A-matrix, potentially leading to performance degradation.
2. HyNoIse is primarily a character-level noise and may not reflect real-world, high-complexity datasets. Is it possible for the authors to provide analysis of the performance in a real task?
3. Missing hyperparameter analysis of β.
4. The first paragraph of Section 1 lacks a relevant citation;
5. For the without noise performance in Table 1, why does LoPE improve compared to HydraLoRA, if I understand correctly, LoPE lies mainly in the two stages of fine-tuning the noise data, but why would this improve the Orig dataset as well?

---

> ### Author Rebuttal · Authors · 2025-07-31
>
> Thank you very much for your recognition of the innovation and overall quality of our work. We are very impressed by your meticulous and insightful comments, through which we deeply sense your expectation for us to make this work better. The core of our work that you considered is absolutely correct, and almost all other reviewers agree with you. Below, we summarize and explain your concerns point by point:
>
> ***1. Sharing and fine-tuning the A-matrix in both Stage I and Stage II raises the question of whether harmful information learned during the first stage (from noisy data) might be absorbed into the A-matrix, potentially leading to performance degradation.***
>
> **Response:** Thank you for your valuable comment! Here is our theoretical analysis regarding this question: Based on the further exploration of the knowledge of matrix A in the HydraLoRA paper, we believe that matrix A learns relatively language environment knowledge, which is very basic and universal, and very little affected by noise(a dataset with added noise can be analogized as a new domain dataset). In addition, we have conducted two experiments in the past few days to address your concerns. Firstly, we conducted a comprehensive visualization of the A and B parameters in LoPE to present the actual changes in each part of the process. Due to format limitations of the rebuttal, we will first provide the key visualization results, and we will include all the figures and a complete analysis in the camera-ready version. Our findings show that during the entire two-stage fine-tuning process, **the position of matrix A in the visualization space changes very little**, which is fully consistent with our previous conclusions in the manuscript, and also aligns with the HydraLoRA paper’s conclusion that matrix A represents general knowledge. And we further conducted experiments by freezing the matrix A obtained after the first stage. The table presents our new findings and reveals an interesting phenomenon: we observed a significant drop in accuracy. Through careful investigation, we identified the root cause and are happy to share and discuss it with you as follows:
>
> | Method                 | MMLU  | PIQA  | SIQA  | GSM8K |
> | ---------------------- | ----- | ----- | ----- | ----- |
> | LoPE(Freeze A in Stage I)         | 43.93 | 74.81 | 46.67 | 11.20 |
> | LoPE(Original Table 1) | 44.42 | 76.28 | 49.03 | 13.72 |
>
>
> Our analysis shows that **freezing matrix A immediately after its initialization in the first stage will hinder convergence and reduce performance**. This finding is consistent with observations from the ablation study in "LoRA: Low-Rank Adaptation of Large Language Models", which documented a similar effect. However, our freezing approach does not result in severe negative impacts, as we still have the second stage of training that unfreezes and normally trains matrix A, though freezing A in Stage I does significantly affect the speed of the first stage.
>
> Based on these results, we draw two key conclusions:
>
> First, as noted above, matrix A exhibits stability—its fundamental structure remains largely unchanged throughout training. Furthermore, as demonstrated in HydraLoRA, the general knowledge encoded in matrix A is resistant to noise-induced perturbations.
>
> Second, and perhaps more importantly, our two-stage training paradigm actually enhances the robustness of matrix A. The dynamic learning process does more than just preserve its initial state; it reinforces its representation of noise while maintaining its core knowledge structure over both stages. This indicates that allowing matrix A to adapt during training, rather than freezing it prematurely, is crucial for achieving optimal noise robustness.
> ﻿
>
>
> ***2.HyNoIse is primarily a character-level noise and may not reflect real-world, high-complexity datasets. Is it possible for the authors to provide analysis of the performance in a real task?***
>
> **Response:** Thank you for your valuable comment! As described in Section 3.2.1, HyNoIse represents a comprehensive noise injection method that extends beyond character-level perturbations. It encompasses **character-level, token-level, label-level, and structural-level** typical noise patterns, which comprehensively cover the common anomalies we have surveyed in real-world scenarios, including **continuous noise in the embedding level**, designed to simulate diverse real-world noise conditions.  I understand your concern about the performance of real-world tasks as we do. Therefore, our evaluation employs a carefully curated suite of benchmark datasets: MMLU, PIQA, SIQA, and GSM8K, collectively spanning mathematics, commonsense reasoning, chemistry, history, law, and numerous other domains.  Here are a few examples to illustrate its properties:
> ﻿
>
> - MMLU (Massive Multitask Language Understanding) spans tasks across multiple domains, enabling comprehensive evaluation of models' language understanding capabilities across diverse knowledge areas.
> - PIQA (Physical Interaction QA) is designed to evaluate models' ability to predict physical object interactions and commonsense physical understanding.
> - SIQA (Social Interaction QA) serves as a benchmark for evaluating social commonsense reasoning and interpersonal intelligence.
> - GSM8K provides grade school math problems requiring multi-step mathematical reasoning.
>
> And we also added experiments on the tasks mentioned in the related work as follows:
>
> | Method | MRR  | NDCG@3 | R@10 | R@100 |
> | ------ | ---- | ------ | ---- | ----- |
> | LeCoRE | 51.1 | 48.5   | 73.9 | 89.7  |
> | LoPE   | 54.7 | 50.1   | 75.6 | 91.3  |
>
> The experimental results demonstrate that our method achieves superior performance across all evaluation metrics, indicating the generalizability of LoPE to real tasks.
>
> We believe that these widely adopted benchmarks collectively provide a comprehensive assessment of our model's practical capabilities across different cognitive tasks and knowledge domains. In addition, we will add more performance and corresponding analysis of LoPE in specific downstream natural language processing tasks in the Camera-Ready version. Thanks again!
>
> ***3.Missing hyperparameter analysis of β.***
>
> **Response:** Thank you for your valuable comment! We are deeply sorry that there is an ambiguous notation regarding β in Equation (3)... To clarify, β serves as a normalization factor rather than a hyperparameter. Since the sum of the subsequent coefficients exceeds 1, we introduce β to ensure that
> $$
> \beta(\omega_D+\sum_{i = 1}^{K}(1+\theta_{iD})\omega_i)=1,
> $$
>
> thereby maintaining consistent scaling across all coefficients (the same principle applies to Equation (4)). This normalization ensures that all components adhere to the same scale. We will revise this notation and provide a clearer explanation in the revised manuscript.
>
> ***4. The first paragraph of Section 1 lacks a relevant citation***
>
> **Response:** Thank you for your valuable comment!  We have added the relevant citations to the introduction section as suggested. We greatly appreciate your guidance in this regard.  We will carefully investigate and add other related work in the Camera-Ready version. If you believe there are other important works relevant to our research contributions that should be cited, we would be very grateful for your valuable suggestions. Thanks again!
>
> ***5. For the without noise performance in Table 1, why does LoPE improve compared to HydraLoRA, if I understand correctly, LoPE lies mainly in the two stages of fine-tuning the noise data, but why would this improve the Orig dataset as well?***
>
> **Response:** Thank you for your valuable comment! Our primary motivation centers on handling data containing substantial noise. After completing the noisy data experiments, we recognized that even "original" datasets **inherently contain noise**, albeit potentially at lower levels. As documented in "*Learning from noisy labels with deep neural networks: A survey*" (cited in our manuscript), no real-world dataset achieves perfect cleanliness.
>
> While our framework treats original data as "zero-noise" theoretically for comparison with our artificially noise-injected datasets, this represents a relative rather than an absolute characterization. The reality of large-scale datasets is that complete clean remains unattainable—**an inherent limitation of current data collection and annotation practices**. Consequently, LoPE demonstrates performance improvements even on unprocessed original datasets because it effectively handles the **latent noise present in ostensibly "clean" data**. This observation underscores the practical utility of our approach beyond artificial noise scenarios.
>
> Thank you again for taking the time to review our response. We sincerely hope that our response can solve your concerns and better evaluate the comprehensive value of our work. Thanks again!

---

> ### Comment · Reviewer_CYJM · 2025-08-06
>
> Thanks for your thorough response and the additional experiments. I've gone through your answers, the reviewer discussions, and the new results.
>
> While your clarifications helped address some of my questions, I still have a few lingering concerns.
> The new findings about freezing matrix A after Stage 1 and the resulting performance drop are interesting and align with prior work, reinforcing that adaptation is key for convergence.
> That said, this doesn’t directly confirm whether harmful noise is being absorbed into matrix A. You mention that its "fundamental structure remains largely unchanged" and that it's "resistant to noise-induced perturbations," but these are more qualitative claims. The performance drop in the ablation study could stem from factors other than noise absorption, making it tricky to draw firm conclusions here. A more direct analysis or a stronger ablation study is welcomed.
>
> As for the other points, your responses have cleared those up.

---

> > ### Author Response · Authors · 2025-08-07
> >
> > Thanks for your insightful comment! We are delighted to address your other concerns through the response. Regarding the question about matrix A's performance and whether it absorbs noise, here is our detailed analysis with relevant experimental results:
> >
> > (1) We fully agree with your observation that freezing matrix A after Stage I and the resulting performance drop can't directly confirm whether noise is being absorbed into matrix A. Freezing matrix A in the first stage impacts performance because it hinders convergence, which indeed cannot directly confirm whether harmful noise is absorbed by matrix A. Therefore, we provide more direct experiments and analyses to demonstrate this.
> >
> > (2) Regarding our previous response about "its fundamental structure remains largely unchanged throughout training," we apologize for any misunderstanding caused by our imprecise expression. About the positional changes of matrices A and B in the visualization mentioned in our previous response, here we provide a detailed explanation: Our visualization method examines whether the parameter distributions of matrices A and B show differences in the parameter space, directly demonstrating the changes across all parameters in either A or B. Specifically, we load LoRA model weight files, extract all parameters, apply t-SNE dimensionality reduction, and visualize the distribution of different A and B parameters on a two-dimensional plane.
> >
> > Based on this approach, we devised a comparative experiment in our previous response: **training with and without HyNoIse data in Stage I to observe changes in matrix A parameters, thereby demonstrating whether harmful noise is absorbed into matrix A** (we apologize again for the imprecise description of this comparative experiment in our previous response). We display training logs from one selected layer as an example of changes. Here we present more detailed results (Bold values indicate changes in mean and variance):
> >
> > Matrix A‘s parameters (with HyNoIse)
> >
> > Tensor: base_model.layers.10.lora_A.weight
> >
> >   Data type: torch.bfloat16
> >
> >   Mean: 0.000051
> >
> >   Std: 0.009827
> >
> >   Min: -0.022949
> >
> >   Max: 0.023315
> >
> >   Sample data: [-0.013427734375, -0.0172119140625, -0.0146484375, -0.008056640625, 0.00567626953125]
> >
> > Matrix A‘s parameters (w/o HyNoIse)
> >
> > Tensor: base_model.layers.10.lora_A.weight
> >
> >   Data type: torch.bfloat16
> >
> >   Mean: 0.000049 **(0.000002)**
> >
> >   Std: 0.010437 **(0.00061)**
> >
> >   Min: -0.029785
> >
> >   Max: 0.025513
> >
> >   Sample data: [-0.006927490234375, -0.012939453125, -0.0101318359375, -0.01226806640625, 0.012939453125]
> >
> > Matrix B‘s parameters (with HyNoIse)
> >
> > Tensor: base_model.layers.10.lora_B0.weight
> >
> >   Data type: torch.bfloat16
> >
> >   Mean: -0.000056
> >
> >   Std: 0.003189
> >
> >   Min: -0.012024
> >
> >   Max: 0.011902
> >
> >   Sample data: [-0.00183868408203125, -0.0036468505859375, 0.0019073486328125, -0.00099945068359375, -0.00127410888671875]
> >
> > Matrix B‘s parameters (w/o HyNoIse)
> >
> > Tensor: base_model.layers.10.lora_B0.weight
> >
> >   Data type: torch.bfloat16
> >
> >   Mean: -0.000017 **(0.000039)**
> >
> >   Std: 0.002243 **(0.007248)**
> >
> >   Min: -0.008423
> >
> >   Max: 0.009888
> >
> >   Sample data: [-0.001495361328125, -0.00136566162109375, 0.00154876708984375, -0.00168609619140625, 0.00067901611328125]
> >
> > We calculated the mean and variation of the parameters (absolute value) of the A and B matrices in all layers, as shown in the following table:
> >
> > |      Module      | Parameter Average | Change value | Change rate |
> > | :--------------: | :---------------: | :----------: | :---------: |
> > | A (with HyNoIse) |     0.000055      |      -       |      -      |
> > | A (w/o HyNoIse)  |     0.000059      |   0.000004   |  original   |
> > | B (with HyNoIse) |     0.000014      |      -       |      -      |
> > | B (w/o HyNoIse)  |     0.000036      |   0.000022   |  **550%**   |
> >
> > As shown, **the average parameter change of matrix A with or without HyNoIse is much smaller than that of matrix B under the same operations**. We consider this demonstrates to some extent that matrix A primarily captures universal contextual features present in all data, with noise influence being substantially smaller than on matrix B. This also aligns with existing research (including HydraLoRA (NIPS) and TeamLoRA (ACL)), which explains asymmetric structures as: A captures general features while B captures specific features, as cited in lines 60-62 of our manuscript.
> >
> > (3) Additionally, a small aspect is that even if we can't guarantee that A is completely unaffected by noisy data in Stage I, matrix A still participates in normal fine-tuning with clean data in Stage II, which further reduces noise influence and ensures proper training before final deployment.
> >
> > In summary, we sincerely appreciate your profound insights and valuable questions! We will incorporate the above comprehensive experimental results and visualization figures, along with corresponding in-depth analyses, in the camera-ready version. Thanks for your recognition of the value of our work and for your constructive feedback! Thanks again.

---

> > > ### Comment · Reviewer_CYJM · 2025-08-09
> > >
> > > Thank you for the additional clarification.

---

> > > > ### Author Response · Authors · 2025-08-09
> > > >
> > > > Thanks again for your tremendous effort. We truly appreciate your valuable comments, which have significantly enhanced the quality of our work, and we are honored to have had the opportunity to address your concerns.

---

### Official Review · Reviewer_vbzw · 2025-07-02

**Clarity:** 2
**Significance:** 3
**Originality:** 3
**Rating:** 4
**Confidence:** 3

**Summary:**

This paper presents a method called LoPE for making large language models more robust to noisy data during fine-tuning. LoPE is designed based on HydraLoRA which is an existing framework that uses asymmetric LoRA adapters. The key idea is to assign a poisoning expert to learn from noisy data. The training includes two steps: 1) the poisoning expert is trained using mixed types of noise, while the other adapters are kept unchanged; 2) the normal adapters are trained on clean data while keeping the poisoning expert frozen. When making predictions, the poisoning expert is turned off so that only the clean experts affect the results. The authors show experimental results on several tasks like MMLU and QA to support their claims.

**Questions:**

See weaknesses

**Ethical Concerns:**

["NO or VERY MINOR ethics concerns only"]

**Final Justification:**

Authors's responses have clarified my major concerns about the method design. I find the proposed method interesting and insightful, and thus lean toward acceptance.

**Limitations:**

Yes

**Quality:**

3

**Strengths And Weaknesses:**

Strengths:
1. The experiments employ state-of-the-art LLMs as backbones.
2. Ablation studies are conducted to understand the effectiveness of each module.

Weaknesses:
1. The problem setting is somewhat unclear. Based on the descriptions in the Introduction and Experiment sections, my understanding is that the fine-tuning data (training data) are assumed to be noisy—for example, by manually injecting 5% noise to simulate real-world scenarios. However, LoPE appears to use both noisy data (in stage 1) and clean data (in stage 2) during training. Since the noisy data are synthetically generated from the clean data using specific strategies in HyNoIse, the actual input to LoPE is still the clean dataset. This makes the problem setup and the role of noise in the training pipeline somewhat confusing.
2. The overall framework is largely built on HydraLoRA. While the extension to noisy data training is appreciated, the core technical contribution is a bit incremental.
3. The results reported for HydraLoRA on MMLU (46.10 at rank r=4) are not representative of the best-known baseline performance. The HydraLoRA paper reports 47.22 at r=8, which makes the performance comparison in Table 1 of this submission less convincing.
4. The motivation and effectiveness of the two-stage fine-tuning strategy are not well explained. In particular, it is unclear why the "poisoning expert" B_D​, which is trained to handle noise, is ignored during inference. Is it because you assume the test data are relatively clean?
5. The paper criticizes prior methods for being limited to specific noise types, yet the HyNoIse strategy also relies on predefined types of discrete and continuous noise. As such, it does not effectively address the core issue it raises. More evidence is needed to support the claim that LoPE generalizes across diverse noise patterns.
6. All compared baselines are general LoRA variants, none of which are explicitly designed or adapted for noisy data. This weakens the empirical claims.
7. The presentation of the paper requires significant improvement. It contains several unclear expressions and awkward phrasing. Some phrases like “generate output with relatively clean knowledge” are vague. A thorough language and writing revision is strongly recommended to improve readability and ensure the technical content is clearly communicated.

---

> ### Author Rebuttal · Authors · 2025-07-31
>
> Thanks for your time and feedback, we deeply sense your expectation for us to make this work better. In this paper, we boldly explored the 'Noise Robustness Only Through Noise' approach and innovatively proposed LoPE. By combining the advantages of multi-stage differentiated training under the PEFT framework, we completely eliminate the requirement of data cleaning. Furthermore, the proposed LoPE can be widely applied to the parameter-efficient fine-tuning of large language models, enhancing its potential to improve model performance with limited computational resources.
>
> Additionally, we consider that the proposed poisoning expert elimination strategy is not only applicable to noise-related tasks but also offers new insights for diverse scenario tasks that are challenging to model or solve directly.
>
> In the main aspects, including motivation, paper quality, experimental results, and reproducibility, our work was fully approved by all other reviewers.
>
> Through your comments, we noticed that there are some misunderstandings about our work. In the following, we will summarize and explain your concerns point by point in order to clarify them.
>
> **R1:**
>
> Thanks for your comment. In our experiments, Orig and Nois represent two independent experimental workflows, not stages within a single pipeline. Each follows the same full two-stage fine-tuning and evaluation process. Specifically, Orig is the original dataset (with inherent noise), while Nois augments it with 5% additional discrete noise to create a more challenging and comparable setting. The clean data (in stage 2) you mentioned actually corresponds to the datasets used in each setting.
> In contrast, the Stage 1 data generated via HyNoIse is additional noise specifically crafted to train the poisoning expert, which is distinct from the Nois setting created for a more challenging and comparable evaluation.
>
> Whether fine-tuning on Orig or Nois, we follow the same mechanism:
>
> Stage I: We use the fine-tuning dataset augmented via HyNoIse to exclusively train only the poisoning expert, while freezing other experts, ensuring that noise adaptation is only highly concentrated in it.
>
> Stage II: We freeze the poisoning expert to preserve its noise adaptation and train the normal experts and router on the dataset without further HyNoIse (i.e., the Orig or Nois dataset itself). Consequently, the learned routing distribution increasingly assigns noisy samples to the poisoning expert and clean samples to others, achieving our intended noise isolation.
>
> Crucially, we use the HyNoIse only in Stage I to train the poisoning expert; it's never used in Stage II. In the inference stage, we mask the poisoning expert, so the final output is generated only from the normal experts.
>
> Regarding "Since the noisy data that synthetically generated from the clean data, and many studies on noise injection or data augmentation rely on clean data to synthesize noise, we consider that this does not diminish their effectiveness. “Data Augmentation Approaches in Natural Language Processing: A Survey” explicitly states that “adding noise to the original data while preserving label validity can improve model robustness.”
>
> Thanks again for your comment! Please feel free to reach out for any further discussion.
>
> **R2:**
>
> Thanks for your comment. With PEFT paradigms becoming the mainstream framework for large language models, we recognize that HydraLoRA serves as the architectural foundation for many subsequent innovations, and we believe that building upon this architecture through novel designs should be encouraged, similar to how many works build upon the LoRA framework. Several of HydraLoRA-based works have been widely acknowledged by the community (e.g., TeamLoRA, ACL 2025). As mentioned below, our core contribution lies in proposing a novel, complementarity-inspired paradigm. We believe that it is not only applicable to noise-related tasks but also provides new insights for more scenarios that are difficult to model or solve directly.
>
> **R3:**
>
> Thanks for your comment. As demonstrated in the experiments of the HydraLoRA paper, higher ranks generally lead to better performance. However, increasing the rank also incurs higher parameter and computational costs, making it necessary to balance scalability and effectiveness. We believe that a similar trend holds for our LoPE framework.
>
> We have added results for LoPE with ranks 8 and 16, and compared them with HydraLoRA at the same ranks. The observed performance trends remain consistent with our previous conclusions, and we also expect a similar trend to hold at higher ranks. † denotes methods fine-tuned on Nois.
>
> | Methods           | MMLU  | PIQA  | SIQA  | GSM8K |
> | ----------------- | ----- | ----- | ----- | ----- |
> | HydraLoRA(r=8) †  | 43.98 | 75.30 | 48.12 | 12.44 |
> | LoPE(r=8) †       | 44.82 | 76.83 | 49.90 | 14.31 |
> | HydraLoRA(r=16) † | 45.20 | 76.92 | 49.63 | 12.95 |
> | LoPE(r=16) †      | 45.76 | 77.22 | 50.87 | 15.01 |
>
> Notably, HydraLoRA’s MMLU result (47.22%) was obtained using Databricks-Dolly-15k. However, we can't find a usable version in their repository. All of our experiments were fine-tuned on the Alpaca-52K dataset (see Section 4.1). We believe that differences in fine-tuning datasets naturally lead to variations in inference performance.
>
> We will include both the results for ranks 4 and 8 as the main results in the camera-ready version, and provide a complete ablation study with corresponding analysis for different ranks in a later section. We thank you again for your valuable suggestion.
>
> **R4:**
>
> Thank you for your comment. The poisoning expert’s core role is to leverage the complementary set concept to facilitate routing, rather than to contribute to output during inference. Please review the process details in R1. We deliberately mask the poisoning expert during inference because it acquires noise-related bias during Stage I, so its outputs inherently carry noise-related bias. If not masked, the model will be affected by the incorrect knowledge learned from noise, which in turn reduces the model's performance. This phenomenon is validated in Section 5.2 of the manuscript (Table 4 and lines 330).
>
> We also believe that LLMs during training typically rely on large-scale datasets of varying quality, inevitably containing noise, whereas the testing phase usually employs high-quality data for specific tasks. This fact is well supported by “Learning from noisy labels with deep neural networks: A survey.” We will include this reference in the camera-ready version. Thanks again!
>
> **R5:**
>
> We clarify our statement in line 42 as follows in our paper. The term “noise type” refers to the scenario-specific noise patterns and characteristics of each dataset in the context of its corresponding task, rather than to a classification of noise as discrete or continuous.
>
> In contrast, LoPE is not tailored to any specific downstream task or domain, and can thus be widely applied to the parameter-efficient fine-tuning of large language models. In our main results, LoPE has already been validated on over 60 diverse NLP tasks, demonstrating its broad applicability. Its design does not rely on any fixed or pre-defined noise type; instead, it incorporates noise at the character, token, label, and structural levels—covering common real-world perturbations (Section 3.2.1).
>
> Additionally, we have included a dedicated comparison experiment against a task-specific denoising model to further evaluate LoPE’s generality in downstream applications. We have conducted an investigation of the representative denoising method LeCoRE (specifically designed for conversational search tasks). We followed their evaluation protocol and conducted experiments under comparable settings. The experimental results are presented below:
>
> | Method | MRR  | NDCG@3 | R@10 | R@100 |
> | ------ | ---- | ------ | ---- | ----- |
> | LeCoRE | 51.1 | 48.5   | 73.9 | 89.7  |
> | LoPE   | 54.7 | 50.1   | 75.6 | 91.3  |
>
> **R6:**
>
> Thank you for your valuable comment! Lope is a general fine-tuning architecture widely applicable to various noise environments and data types. We consider that it does not conflict with other denoising methods and can be widely combined with them. Therefore, we primarily compared it with other PEFT methods of the same category.
>
> Despite the above discussion, we fully agree with your point and have conducted an investigation of the two representative denoising methods. The experimental results are presented in R5. We will include more comparative experimental results and corresponding analyses in the camera-ready version.
>
> **R7:**
>
> Thank you for your comment. Regarding the phrase “generate output with relatively clean knowledge” in the caption of Figure 2, although we hope that the model can produce clean outputs, in practice, it is difficult for the model to consistently generate perfectly clean and ideal results, which is why we avoided using absolute terms. Nonetheless, we sincerely appreciate your careful review and will refine this expression in the camera-ready version. We will also conduct a thorough review of the entire paper to ensure clarity throughout. Thank you again!
>
> For all typos, grammatical errors, and unclear notations, we will carefully revise them one by one. Despite these writing issues, we sincerely argue for the value of this work. As mentioned earlier, we have completely eliminated the need for manual noise cleaning by adopting the opposite approach of noise injection. Moreover, we believe the proposed poisoning expert mechanism offers broad inspiration for diverse scenario tasks in the future. Based on the feedback from all the other reviewers, our paper is well-worked and of great value to the academic community. Therefore, we warmly welcome continued discussion and exchange, and we sincerely hope you can take the comprehensive value of this work into consideration. Thanks again!

---

> > ### Comment · Reviewer_vbzw · 2025-08-02
> > **Reviewer Response**
> >
> > Thank you for the authors' rebuttal. I would like to adjust my rating since most of my concerns have been clarified.
> >
> > One further question is still relevant to the noise injection part. To simulate data noise in the experiments, you actually used the same HyNoIse module as in the LoPE algorithm (I initially misunderstood this point). This raises a new concern that the good performance of LoPE may heavily rely on that consistency. Considering that real-world noise is often different from simulation and more complex, have you tried:
> >
> > 1) using different noise operations in LoPE and for creating the $Nois$ dataset? (e.g., character-level perturbation in LoPE and token-level perturbation for the $Nois$ dataset), and
> >
> > 2) applying more realistic noise types to create $Nois$, such as replacing a word with commonly misspelled forms (e.g., receive → recieve)?
> >
> > Another question is: how do you confirm that the improvements mainly come from the Poisoning Expert rather than the HyNoIse simulation? Have you tried directly applying HyNoIse to HydraLoRA? Some previous papers [1,2,3] have also demonstrated improved robustness by injecting noise into the input during fine-tuning. While they were primarily designed for full fine-tuning, they should also be applicable to LoRA-based frameworks.
> >
> > [1] HyPe: Better Pre-trained Language Model Fine-tuning with Hidden Representation Perturbation.
> >
> > [2] Better Fine-Tuning by Reducing Representational Collapse.
> >
> > [3] Improving Pre-trained Language Model Fine-tuning with Noise Stability Regularization.

---

> > > ### Author Response · Authors · 2025-08-02
> > >
> > > We sincerely thank you for your careful review of our manuscript and your valuable comments. We greatly appreciate the issues you have raised and are actively conducting additional experiments to further improve and strengthen our study. Since some of these experiments require time for implementation and validation, we are working diligently to complete them and to provide more comprehensive and reliable data support. We sincerely appreciate your understanding and patience.
> > >
> > > Thanks again!

---

> > > ### Author Response · Authors · 2025-08-03
> > >
> > > **Thank you for your feedback and patience. Please allow us to express our most sincere gratitude for your exceptionally thorough and insightful evaluation of our work. Your scholarly rigor, manifested through the meticulous analysis and constructive feedback, has not only profoundly impressed our research team but has also provided invaluable guidance for refining our academic contributions. Here we respond to each of the two questions point by point:**

---

> > > ### Author Response · Authors · 2025-08-03
> > >
> > > ***Q1: One further question is still relevant to the noise injection part. To simulate data noise in the experiments, you actually used the same HyNoIse module as in the LoPE algorithm (I initially misunderstood this point). This raises a new concern that the good performance of LoPE may heavily rely on that consistency. Considering that real-world noise is often different from simulation and more complex, have you tried:***
> > >
> > > ***1.using different noise operations in LoPE and for creating the Nois dataset? (e.g., character-level perturbation in LoPE and token-level perturbation for the Nois dataset), and***
> > >
> > > ***2.applying more realistic noise types to create Nois, such as replacing a word with commonly misspelled forms (e.g., receive → recieve)?***
> > >
> > >
> > >
> > > ***Response:***
> > >
> > > Thanks for your insightful comment. The noise injection method (Hynoise) in LoPE comprises both discrete and continuous noise components. As you mentioned, the discrete noise component indeed shares certain similarities with the Nois dataset construction. However, we sincerely argue for LoPE's effectiveness from the following three perspectives:
> > >
> > > (1) Theoretically, although similar discrete noise operations are employed in both Nois dataset creation and LoPE training, we additionally incorporate **continuous noise injection during LoPE's training phase**. This continuous noise simulates various noise patterns and structures to a certain extent, including s**tructural noise, token-level perturbations, incomplete text segments, random truncations, format issues**, etc., thereby enhancing the model's potential to handle diverse noise patterns.
> > >
> > > (2) Following your suggestions, we immediately conducted experiments with different operations for LoPE training and dataset construction, including (Multi-inconsistent noise injection operation between  **Nois** dataset construction and discrete noise in HyNoIse). The results are presented below; the number in (.) represents the number of types of discrete noise contained:
> > >
> > > |                  Model                  |                       Dataset                        |                           HyNoIse                           |       MMLU        |       PIQA        |       SIQA        | GSM8K             |
> > > | :-------------------------------------: | :--------------------------------------------------: | :---------------------------------------------------------: | :---------------: | :---------------: | :---------------: | ----------------- |
> > > | LoPE **(Inconsistent noise injection)** |   Orig+**noise character insertion** **(1 type)**    |   Continuous noise+**Word order shuffling** **(1 type)**    | **44.58** (+0.34) | **75.15** (+0.12) | **50.11** (+0.36) | **14.92** (+2.14) |
> > > |                HydraLoRA                |   Orig+**noise character insertion** **(1 type)**    |                            None                             |       44.24       |       75.03       |       49.75       | 12.78             |
> > > | LoPE **(Inconsistent noise injection)** |      Orig+**word order shuffling** **(1 type)**      |       Continuous noise+**Character deletion(1 type)**       | **45.09** (+0.72) | **75.84** (+0.63) | **50.55** (+0.95) | **14.64** (+2.3)  |
> > > |                HydraLoRA                |      Orig+**word order shuffling** **(1 type)**      |                            None                             |       44.37       |       75.21       |       49.65       | 12.34             |
> > > | LoPE **(Inconsistent noise injection)** | Orig+**word replacement** **(1 type you mentioned)** | Continuous noise+**Noise character insertion** **(1 type)** | **45.13** (+0.82) | **75.35** (+0.28) | **50.17** (+0.44) | **14.73**(+2.34) |
> > > |                HydraLoRA                | Orig+**word replacement** **(1 type you mentioned)** |                            None                             |       44.31       |       75.07       |       49.73       | 12.39             |
> > >
> > > We can observe that when different operations are applied for LoPE training and dataset construction (each row in the table represents a distinct operation), compared to the baseline HydraLoRA, all LoPE experiments maintained considerable improvements. This demonstrates LoPE's generalization capability across different noise categories to a certain extent.
> > >
> > > (3) Another small aspect is that our method achieves considerable performance improvements on ***orig*** datasets without additional noise injection (as shown in the first two rows of Table 1 in the paper). Since real-world datasets **inherently contain various types of noise**, we consider that it can further demonstrate the effectiveness and generalizability of LoPE on naturally occurring noisy data to a certain extent.
> > >
> > > We sincerely appreciate your profound insights and valuable questions. We will incorporate these comprehensive experimental results along with corresponding in-depth analyses in the camera-ready version. Thank you again for your constructive feedback!

---

> > > ### Author Response · Authors · 2025-08-03
> > >
> > > ***Q2: Another question is: how do you confirm that the improvements mainly come from the Poisoning Expert rather than the HyNoIse simulation? Have you tried directly applying HyNoIse to HydraLoRA? Some previous papers [1,2,3] have also demonstrated improved robustness by injecting noise into the input during fine-tuning. While they were primarily designed for full fine-tuning, they should also be applicable to LoRA-based frameworks.***
> > >
> > > ***[1] HyPe: Better Pre-trained Language Model Fine-tuning with Hidden Representation Perturbation.***
> > >
> > > ***[2] Better Fine-Tuning by Reducing Representational Collapse.***
> > >
> > > ***[3] Improving Pre-trained Language Model Fine-tuning with Noise Stability Regularization.***
> > >
> > >
> > >
> > > ***Response:***
> > >
> > > Thank you for your insightful comment. Following your suggestion, we also conducted additional experiments by directly applying HyNoIse to the original HydraLoRA framework, and the results are presented below:
> > >
> > > | Methods                             | MMLU      | PIQA      | SIQA      | GSM8K     |
> > > | ----------------------------------- | --------- | --------- | --------- | --------- |
> > > | HydraLoRA† **(with HyNoIse)**       | 40.57     | 73.40     | 45.03     | 8.47      |
> > > | HydraLoRA† (*Table 1 in the paper*) | 43.08     | 74.92     | 47.29     | 11.83     |
> > > | **LoPE†** (*Table 1 in the paper*)  | **44.42** | **76.28** | **49.03** | **13.72** |
> > >
> > > We observe that the experimental results demonstrate a noticeable performance degradation when HyNoIse is applied to the original HydraLoRA architecture (HydraLoRA+HyNoIse). We consider that this is primarily because of the absence of the **division of labor between normal experts and poisoning experts**, as well as the multi-stage **differentiated training**. Without these processes, **noisy and clean data equally affect the model, which confuses the model from the very beginning.**
> > >
> > >
> > >
> > > As a result, we consider that LoPE indeed aims to achieve noise robustness through noise injection; **its key contribution is not in the noise injection itself**, but in **the mechanism and process** within the LoPE architecture, which enables explicit noise filtering through multi-stage differentiated training.
> > >
> > >
> > >
> > > In addition, compared to the related job you recommended, we also found several distinguishing characteristics of LoPE beyond its method innovations:
> > >
> > >
> > >
> > > LoPE addresses different problem scenarios with distinct objectives. Compared to these mentioned methods, LoPE does not require implementation validation of parameter selection and has a certain level of interpretability, applying to various scenarios without increasing computational complexity.
> > >
> > >
> > >
> > > Thanks again for your profound analysis and valuable questions. We will provide an in-depth analysis of these aspects along with the complete experimental results in the camera-ready version.
> > >
> > > **In summary, we are deeply cognizant of the significant effort invested in your comprehensive review, and we fully understand that your clarifications stem from your expectation for our work to reach its full potential! Thank you for your tremendous and valuable efforts to improve it, and we warmly welcome further discussion if you have any other questions. Thanks again!**

---

> > > > ### Comment · Reviewer_vbzw · 2025-08-07
> > > >
> > > > Thank you for your further response. Since my major concerns have been clarified, I will increase my score to borderline accept. Some remaining weaknesses are as follows:
> > > >
> > > > - When simulating data noise, I suggest the authors create the $Nois$ dataset using real-world linguistic noise, such as replacing words with commonly misspelled forms. It would also be beneficial to extract a noisy subset from the $Orig$ dataset using LLM tools.
> > > >
> > > > - Baseline methods such as [1, 2, 3] should be included in the comparison as they are also noise-based methods. A comparison of computational complexity would be also helpful.
> > > >
> > > > Nevertheless, I find the proposed method interesting and insightful, and thus lean toward acceptance.

---

> > > > > ### Author Response · Authors · 2025-08-08
> > > > >
> > > > > We sincerely thank you for your insightful comments and recognition of the value of our work. More importantly, we appreciate the tremendous efforts you have made to further optimize the work.
> > > > >
> > > > > We will follow your suggestion and further enhance our experiments by introducing more real-world linguistic noise in the revised version. In addition, a noisy subset from the Orig dataset will be extracted by LLM tools to simulate practical scenarios better.
> > > > >
> > > > > Besides, we will incorporate a detailed comparison with the baseline methods you mentioned ([1], [2], [3]) and other relevant studies from both theoretical and experimental perspectives in the camera-ready version.
> > > > >
> > > > > Thanks again for your time and valuable feedback!

---

### Official Review · Reviewer_tqFZ · 2025-07-02

**Clarity:** 3
**Significance:** 3
**Originality:** 3
**Rating:** 5
**Confidence:** 4

**Summary:**

This paper introduces LoPE (asymmetric LoRA poisoning experts), a pretty clever approach to handling noisy data during fine-tuning. Instead of trying to clean noise, they deliberately inject it. They use an asymmetric LoRA structure with a dedicated "poisoning expert" trained to handle noise. During inference, they simply mask this expert out. Their two-stage process is neat: first train the poisoning expert on noisy data, then freeze it while training other components. They ran experiments across several benchmarks and showed some improvements over other PEFT methods.

**Questions:**

See above.

**Ethical Concerns:**

["NO or VERY MINOR ethics concerns only"]

**Final Justification:**

The rebuttal has addressed my concerns.

**Limitations:**

See above.

**Paper Formatting Concerns:**

None.

**Quality:**

3

**Strengths And Weaknesses:**

## Strengths

1. The "noise through noise" approach represents a creative departure from conventional denoising methods.

2. Their use of asymmetric LoRA architecture to separate functionality between experts is technically sound and well-executed.

3. The method shows promising results across multiple tasks, with notable performance on mathematical reasoning and complex reasoning tasks.

4. The DyCompEnSate approach demonstrates careful consideration of the knowledge gaps that could arise from expert masking.

## Weaknesses

1. This paper mostly compares against other PEFT methods like LoRA and HydraLoRA, but I'm missing comparisons with actual denoising methods (LeCoRE, MICL, etc.). I'd like to see them pit their method against mainstream denoising techniques - that would make their case much stronger.

2. This paper primarily tests with 5% noise injection, but then cite research saying real-world noise can be 8%-38.5%.
   - Figure 4 shows different noise ratios but doesn't dig into why performance drops at higher levels.
   - Would love to see how this holds up at 10%, 20%, 30% noise - does it collapse or gracefully degrade?

3. It would be better that include visualization or analysis of its internal representations to describe what exactly is this poisoning expert learning.

---

> ### Author Rebuttal · Authors · 2025-07-31
>
> Thanks for your time and feedback! We admire your deep understanding of this work and your extremely valuable comments. The core of our work that you considered is absolutely correct, and all other reviewers agree with you. Below, we summarize and explain your concerns:
>
> ***1. This paper mostly compares against other PEFT methods like LoRA and HydraLoRA, but I'm missing comparisons with actual denoising methods (LeCoRE, MICL, etc.). I'd like to see them pit their method against mainstream denoising techniques - that would make their case much stronger.***
>
> **Q1 Response:** Thank you for your valuable comment! In general, aside from the performance improvement, our main advantage is that we completely eliminate the need for data cleaning. In contrast, by adding noise at a low cost, we achieve the goal of "Noise Robustness Through Noise." This not only overcomes issues like uneven data distribution caused by data cleaning but also provides broad inspiration for diverse scenario tasks that are challenging to model or solve directly. Since it is a general fine-tuning architecture widely applicable to various noise environments and data types, we consider that it does not conflict with other denoising methods and can be widely combined with them. Therefore, we primarily compared it with other PEFT methods of the same category.
>
> Despite the above discussion, we fully agree with your point and have conducted an investigation of the two representative denoising methods you recommended: LeCoRE and MICL. For LeCoRE, a method specifically designed for conversational search tasks, we followed their evaluation protocol and conducted experiments under comparable settings. The experimental results are presented below:
>
>
>
> | Method | MRR  | NDCG@3 | R@10 | R@100 |
> | ------ | ---- | ------ | ---- | ----- |
> | LeCoRE | 51.1 | 48.5   | 73.9 | 89.7  |
> | LoPE   | 54.7 | 50.1   | 75.6 | 91.3  |
>
> The experimental results demonstrate that our method achieves superior performance across all evaluation metrics, indicating the generalizability of LoPE to domain-specific tasks. As MICL is a multi-modal model that presents challenges for direct comparison and the author does not have open-source code, we have not included its results at this stage. We commit to conducting comprehensive and scientifically rigorous evaluations of various denoising models across different scenarios, including LeCoRE and MICL, and will present the corresponding results and analyses in the camera-ready version. Thanks again!
>
>
>
> ***2. This paper primarily tests with 5% noise injection, but then cite research saying real-world noise can be 8%-38.5%.***
>
> **Q2 Response:** Thanks for your comment! As mentioned in Chapter 4.2 of our original text, given that the original dataset already contains some inherent noise, we used an additional rate of 5% to create a more challenging environment, where they work together with the inherent noise in the dataset and collectively affect the model.
>
>
> **2.1 Figure 4 shows different noise ratios but doesn't dig into why performance drops at higher levels.**
>
> **Q2.1 Response:** Due to space limitations, we provided only a brief discussion of this phenomenon. Here, we present a more comprehensive analysis of the observed performance drop under high levels of HyNoIse.
>
> Our analysis identifies several key contributing factors:
>
> **(1) Negative Effects of Excessive Noise:** When the HyNoIse ratio exceeds 10%, we observe a significant drop in performance. This is because excessive noise impairs the router’s ability to **distinguish between clean and noisy data**, leading to confusion in expert weight allocation and ultimately weakening the model's discriminative capacity to a certain extent.
>
> **(2) Distortion of Data Distribution:** As the HyNoIse ratio increases, the statistical properties of noisy samples can overwhelm and potentially disrupt the underlying distribution patterns of the original data.
>
> **(3) Randomization of Weight Allocation:** As the HyNoIse level rises, the embedding vectors of noisy samples increasingly overlap with those of clean data. This expanded overlap prevents the router from effectively separating the “clean” and “noisy” paths, resulting in an almost uniform distribution of expert weights. Consequently, the originally designated "poisoning expert" (BD) is forced to process clean data, while the normal experts (e.g., B1/B2) are exposed to noisy samples, undermining their role as independent repositories of "clean knowledge."
>
> To aid understanding, we offer an analogy (imperfect, but conceptually helpful):
>
> **Low HyNoIse:** Similar to giving the model a “**vaccine**” — a small dose of virus (noise) that stimulates immunity (robustness).
>
> **High HyNoIse:** Comparable to injecting a large amount of **virus** directly, overwhelming the immune system (model) and compromising normal functionality.
>
>
>
> **2.2 Would love to see how this holds up at 10%, 20%, 30% noise - does it collapse or gracefully degrade?**
>
> **Q2.2 Response:** Thank you for your interest! We have conducted extensive supplementary experiments in this regard and discovered some interesting phenomena. We systematically evaluated noise ratios of **3.5%, 5%, 8%, 10%, 20%, and 30%** in the noisy data. Partial results are shown below.
>
> | Noise Ratio in Nois                 | MMLU  | PIQA+SIQA                              | gsm8k |
> | ----------------------------------- | ----- | -------------------------------------- | ----- |
> | 3.5%                                | 44.59 | 63.31(*Table 2 in the original paper*) | 15.31 |
> | 5%(*Table 1 in the original paper*) | 44.42 | 62.66                                  | 13.72 |
> | 8%                                  | 44.15 | 61.86(*Table 2 in the original paper*) | 11.83 |
> | 10%                                 | 43.81 | 61.27                                  | 9.25  |
> | 20%                                 | 42.35 | 60.14                                  | 8.04  |
> | 30%                                 | 40.78 | 59.50                                  | 7.60  |
> | 30%(*HydraLoRA*)                  | 37.96 | 56.83                                  | 7.12  |
>
> From the results, it is evident that while overall performance does decline with increasing noise levels, **it does not completely collapse**.
> We analyze the following reasons:
>
> **(1)** Our method delays performance collapse under higher noise conditions.
>
> **(2)** The overall quality of fine-tuning datasets impacts final inference performance. When the noise reaches a certain level, it blurs the boundary between clean data and noise, and fine-tuning becomes harmful, highlighting the importance of fine-tuning data quality.
>
> **(3)** When data noise rises to a certain degree, the accuracy decline rate for mathematics datasets requiring high precision actually slows down. This may be because mathematical pattern-related content in the fine-tuning dataset has already been corrupted, resulting in less impact on the model itself (the model may no longer recognize that it is learning mathematics-related knowledge).
>
> In the camera-ready version, we will present the complete set of these supplementary experiments to provide a comprehensive view of our work and enrich the research community's understanding of LLM data noise.  We will include comprehensive results across various noise ratio settings in the camera-ready version, along with an in-depth analysis of the observed performance variations and the underlying differences between real-world noise and synthetically injected training noise in the corresponding section.
>
>
>
> ***3. It would be better that include visualization or analysis of its internal representations to describe what exactly is this poisoning expert learning.***
>
> **Q3 Response:** Thank you for your valuable comment! Following your advice, we have conducted a comprehensive visualization of the A and B parameters in LoPE, which follows the LoRA matrix visualization method released in the HydraLoRA GitHub repository.  Although we cannot provide visualizations here due to the format limitation of the rebuttal, we list the key data and conclusions here. Overall, our findings show that **during the entire two-stage fine-tuning process, the position of matrix A in the visualization space changes very little**, which is fully consistent with our previous conclusions in the manuscript, and also aligns with the HydraLoRA paper’s conclusion that matrix A represents general knowledge.
> In contrast, we observed significant positional changes between the standard expert B matrix and the poisoning B matrix across the training stage. We attribute this phenomenon to the B matrix capturing knowledge contaminated by noise, which is fundamentally **different from the clean knowledge encoded by other experts**.  This aligns with our hypothesis that the poisoning expert effectively diverts tokens affected by noise. We will add the complete visualization process and content mentioned above in the camera-ready version and provide corresponding theoretical analysis and explanations. Thank you again for your insightful suggestion!
>
>
>
> Thank you again for taking the time to review our response. We sincerely hope that our response can solve your concerns and better evaluate the comprehensive value of our work. Thanks again!

---

> > ### Comment · Reviewer_tqFZ · 2025-08-05
> >
> > Thank you for your detailed rebuttal that addresses my concerns. I have raised my score.

---

> > > ### Author Response · Authors · 2025-08-05
> > >
> > > We sincerely appreciate your profound understanding of our work, as well as the time and effort you generously devoted to helping us refine it further.
> > >
> > > Thank you again!

---

### Note · Authors · 2025-08-13

We thank the Area Chair for taking the time to review our replies and manuscript! We also thank the reviewers for their valuable comments and recognition of our work! Following the additional experiments and analyses, we carefully addressed and resolved every concern and suggestion raised by the reviewers, treating each as an opportunity for further learning and improvement.

For a long time, existing methods for improving noise robustness have often relied on varying degrees of data cleaning and preprocessing. This not only significantly increases costs but also introduces limitations such as subjectivity and uneven data distribution. In contrast, injecting noise is a much more cost-effective and objective automated process.

Based on this, we boldly explored the 'Noise Robustness Through Noise' approach and innovatively proposed LoPE. By combining the advantages of the asymmetric LoRA MoE architecture, we were able to enhance the model's noise robustness purely through noise injection. This method eliminates the requirement of data cleaning, thereby achieving noise robustness at a low cost. Furthermore, the proposed LoPE can be widely applied to the parameter-efficient fine-tuning of LLMs, greatly enhancing its potential to improve model performance with limited computational resources.

Additionally, we consider that the proposed poisoning expert elimination strategy is not only applicable to noise-related tasks but also offers new insights for diverse scenario tasks that are challenging to model or solve directly.

In the main aspects, including motivation, paper quality, and experimental results, our work was fully approved by all reviewers. Once again, we sincerely thank all the reviewers and the Area Chair for their time, effort, and constructive feedback, which have greatly contributed to further improving the quality of our work.

---

### Decision · Program_Chairs · 2025-09-17

**Decision:**

Accept (poster)

**Comment:**

In this paper the authors introduce LoPE, a novel method for robust fine-tuning on noisy data using a dedicated "poisoning expert" within an asymmetric LoRA architecture. The idea is that during training, since the poisoning expert adds noise, other experts are encouraged to generate nose resistant representations which during inferencing become slightly stronger since the poison expert is masked. This is an interesting approach in my opinion. Reviewers praised this "noise through noise" approach as creative, technically sound, and promising. There were concerns about the lack of comparison with dedicated denoising methods, limited testing at higher noise ratios, an unclear problem setup, and about the method's success being dependent the same synthetic noise for both training and evaluation.

Authors, in their response, while addressing the concerns, have made clarifications, presented new experiments showing LoPE outperforms a standard denoising method and degrades gracefully at noise levels up to 30%, and show that inconsistent noise types to prove the method's robustness.

Overall this paper will be of interest to the community.